# MODEL-TARGETED POISONING ATTACKS WITH PROVABLE CONVERGENCE

## ABSTRACT

In a poisoning attack, an adversary with control over a small fraction of the training data attempts to select that data in a way that induces a model that misbehaves in a particular way desired by the adversary, such as misclassifying certain inputs. We propose an efficient poisoning attack that can target a desired model based on online convex optimization. Unlike previous model-targeted poisoning attacks, our attack comes with provable convergence to *any* achievable target classifier. The distance from the induced classifier to the target classifier is inversely proportional to the square root of the number of poisoning points. We also provide a lower bound on the minimum number of poisoning points needed to achieve a given target classifier. Our attack is the first model-targeted poisoning attack that provides provable convergence, and in our experiments it either exceeds or matches the best state-of-the-art attacks in terms of attack success rate and distance to the target model. In addition, as an online attack our attack can incrementally determine nearly optimal poisoning points.

## 1 INTRODUCTION

State-of-the-art machine learning models require a large amount of labeled training data, which often depends on collecting data and labels from untrusted sources. A typical application is email spam filtering, where a spam detector filters out spam messages based on features (e.g., presence of certain words) and periodically updates the model based on newly received emails labeled by users. In such a setting, spammers can generate "non-spam" messages by injecting non-related words or benign words, and when models are trained on these "non-spam" messages, the filtering accuracy will drop significantly (Lowd & Meek, 2005). Such attacks are known as *poisoning attacks*, and a training process that collects labels or data from untrusted sources is potentially vulnerable to them.

Poisoning attacks can be categorized into *objective-driven attacks* and *model-targeted attacks* depending on whether a target model is considered in the attack process. Objective-driven attacks have a specific attacker objective and aim to achieve the attack objective by generating the poisoning points; model-targeted attacks have a specific target classifier in mind and aim to induce that target classifier by generating a minimal number of poisoning points. Objective-driven attacks are most commonly studied in the existing literature. The attacker objective is typically one of two extremes: *indiscriminate* attacks, where the adversary's goal is simply to decrease the overall accuracy of the model (Biggio et al., 2012; Xiao et al., 2012; Mei & Zhu, 2015b; Steinhardt et al., 2017; Koh et al., 2018); and *instance-targeted* attacks, where the goal is to produce a classifier that misclassifies a particular known input (Shafahi et al., 2018; Zhu et al., 2019; Koh & Liang, 2017). Recently, Jagielski et al. (2019) introduced a more realistic attacker objective known as a *subpopulation* attack, where the goal is to increase the error rate or obtain a particular output for a defined subpopulation of the data distribution. Attacker objectives for realistic attacks are diverse and designing a unified and effective attack strategy for different attacker objectives is hard. Gradient-based local optimization is most commonly used to construct poisoning points for a particular attacker objective (Biggio et al., 2012; Xiao et al., 2012; Mei & Zhu, 2015b; Koh & Liang, 2017; Shafahi et al., 2018; Zhu et al., 2019). Although these attacks can be modified to fit other attacker objectives, since they are based on local optimization techniques they can easily get stuck into bad local optima and fail to find effective sets of poisoning points (Steinhardt et al., 2017; Koh et al., 2018). To circumvent the issue of local optima, Steinhardt et al. (2017) formulate an indiscriminate attack as a min-max optimization problem and

solve it efficiently using online convex optimization techniques. However, the strong min-max attack only applies to the indiscriminate setting.

In contrast, *model-targeted attacks* incorporate the attacker objective into a target model and hence, the target model can reflect any attacker objective. Thus, the same model-targeted attack methods can be directly applied to a range of indiscriminate and subpopulation attacks just by finding a suitable target model. Mei & Zhu (2015b) first introduced a target model into a poisoning attack, but their attack is still based on gradient-based local optimization techniques and suffers from bad local optima (Steinhardt et al., 2017; Koh et al., 2018). Koh et al. (2018) proposed the KKT attack, which converts the complicated bi-level optimization into a simple convex optimization problem utilizing the KKT condition, avoiding the local optima issues. However, their attack only works for margin based losses and does not provide any guarantee on the number of poisoning points required to converge to the target classifier.

In this work, we focus on model-targeted attacks and aim to understand the feasibility of a poisoning adversary to induce any target model. In particular, we find both theoretical and empirical bounds on the sufficient (and necessary) number of poisoning points to get close to a specific target classier. [1]

**Contributions.** Our main contributions involve developing a principled and general model-targeted poisoning attack strategy, along with a proof that the model it induces converges to the target model. Our poisoning method takes as input a target model, and produces a set of poisoning points. We prove that the model induced by training on the original training data with these points added, converges to the target classifier as the number of poison points increases, given that the loss function is convex and proper regularization is adopted in training (Theorem 4.1). Previous model-targeted attacks lack of such convergence guarantees. We then prove a lower bound on the minimum number of poisoning points needed to reach the target model (Theorem 4.2), given that the loss function for empirical risk minimization is convex. Such a lower bound can be used to estimate the optimality of model-targeted poisoning attacks and also indicate the intrinsic hardness of attacking different targets. Our attack is also efficient in incremental poisoning scenario as it works in an online fashion and can incrementally find poisoning points that are nearly optimal. Previous model-targeted attacks work with fixed number of poisoning points and need to know the poisoning budget in advance. We run experiments to compare our attack to the state-of-the-art model-targeted attack (Koh et al., 2018). We first evaluate the convergence our attack to the target model and find that, under same number of poisoning points, classifiers induced by our attack are closer to the target models than the best known attack, for all the target classifiers we tried. Then, we evaluate the success rate of our attack, and find that it has superior performance than the state-of-the-art in the more realistic subpopulation attack scenario, and comparable performance in the conventional indiscriminate attack scenario (Section 5).

## 2    PROBLEM SETUP

The poisoning attack proposed in this paper applies to multi-class prediction tasks or regression problems (by treating the response variable as an additional data feature), but for simplicity of presentation we consider a binary prediction task, $h : \mathcal{X} \to \mathcal{Y}$, where $X \subseteq \mathbb{R}^d$ and $\mathcal{Y} = \{+1, -1\}$. The prediction model $h$ is characterized by parameters $\theta \in \Theta \subseteq \mathbb{R}^d$. We define the non-negative convex loss on an individual point, $(x, y)$, as $l(\theta; x, y)$ (e.g., hinge loss for SVM model). We also define the empirical loss over a set of points $A$ as $L(\theta; A) = \sum_{(x,y) \in A} l(\theta; x, y)$.

We adopt the game-theoretic formalization of the poisoning attack process from Steinhardt et al. (2017) to describe our model-targeted attack scenario:

1. $N$ data points are drawn uniformly at random from the true data distribution over $\mathcal{X} \times \mathcal{Y}$ and form the clean training set, $\mathcal{D}_c$.
2. The adversary, with knowledge of $\mathcal{D}_c$, the model training process and the model space $\Theta$, generates a target classifier $\theta_p \in \Theta$ that satisfies the attack goal.
3. The adversary produces a set of poisoning points, $\mathcal{D}_p$, with the knowledge of $\mathcal{D}_c$, model training process, $\Theta$ and $\theta_p$.
4. Model builder trains the model on $\mathcal{D}_c \cup \mathcal{D}_p$ and produces a classifier, $\theta_{atk}$.

---

[1]Similar to previous works, in this paper, we focus on designing a model-targeted attack that works for any achievable target model and leave the exploration of finding better target classifiers as the future work.

The adversary's goal is that the induced classifier, $\theta_{atk}$, is close to the desired target classifier, $\theta_p$ (Section 4.2 discusses how this distance is measured). Step 2 corresponds to the target classifier generation process. Our attack works for any target classifier, and in the paper we do not focus on the question of how to find the best target classifier to achieve a particular adversarial goal but simply adopt the heuristic target classifier generation process from Koh et al. (2018). Step 3 corresponds to our model-targeted poisoning attack, and is also the main contribution of the paper.

We assume the model builder trains a model through empirical risk minimization (ERM) and the training process details are known to the attacker:

$$\theta_c = \underset{\theta \in \Theta}{\arg\min} \frac{1}{|\mathcal{D}_c|} L(\theta; \mathcal{D}_c) + C_R \cdot R(\theta) \tag{1}$$

where $R(\theta)$ is the regularization function (e.g., $\frac{1}{2}\|\theta\|_2^2$ for SVM model).

**Threat Model.** We assume an adversary with full knowledge of training data, model space and training process. Although this may be unrealistic for many scenarios, this setting allows us to focus on a particular aspect of poisoning attacks, and is the setting used in many prior works (Biggio et al., 2011; Mei & Zhu, 2015b; Steinhardt et al., 2017; Koh et al., 2018; Shafahi et al., 2018). We assume an addition-only attack where the attacker only adds poisoning points into the clean training set. A stronger attacker may be able to modify or remove existing points, but this typically requires administrative access to the system. The added points are unconstrained, other than being value elements of the input space. They can have arbitrary features and labels, which enables us to perform the worst case analysis on the robustness of models against addition-only poisoning attacks. Although some previous works also allow arbitrary selection of the poisoning points (Biggio et al., 2011; Mei & Zhu, 2015b; Steinhardt et al., 2017; Koh et al., 2018), others put different restrictions on the poisoning appoints. A clean-label attack assumes adversaries can only perturb the features of the data, but the label is given by an oracle labeler (Koh & Liang, 2017; Shafahi et al., 2018; Zhu et al., 2019; Huang et al., 2020). In label-flipping attacks, adversaries are only allowed to change the labels (Biggio et al., 2011; Xiao et al., 2012; 2015; Jagielski et al., 2019). These restricted attacks are weaker than the poisoning attacks without restrictions (Koh et al., 2018; Hong et al., 2020).

## 3 RELATED WORK

The most commonly used poisoning strategy is gradient-based attack. Gradient-based attacks iteratively modify a candidate poisoning point $(\hat{x}, \hat{y})$ in the set $\mathcal{D}_p$ based on the test loss defined on $\hat{x}$ (keeping $\hat{y}$ fixed). This kind of attack was first studied on SVM models (Biggio et al., 2012), and later extended to linear and logistic regression (Mei & Zhu, 2015b), and recently to larger neural network models (Koh & Liang, 2017; Yang et al., 2017; Muñoz-González et al., 2017; Shafahi et al., 2018; Zhu et al., 2019; Huang et al., 2020). Jagielski et al. (2018) also studied gradient attacks and principled defenses on linear regression tasks. Their work studies linear regression while in this paper, we mainly focus on binary classification, although our attack can also be extended to regression tasks. More importantly, our attack aims to induce a target model by generating poisoning points while Jagielski et al. (2018)'s attack tries to increase the Mean Squared Error of the linear regression task with a fixed poisoning budget. In addition to classification and regression tasks, gradient-based poisoning attacks are also applied to topic modeling (Mei & Zhu, 2015a), collaborative filtering (Li et al., 2016) and algorithmic fairness (Solans et al., 2020).

Besides the gradient-based attacks, researchers also utilize generative adversarial networks to craft poisoning points efficiently for larger neural networks, however, the effectiveness of the attack is limited (Yang et al., 2017; Muñoz-González et al., 2019). The strongest attacks so far are the KKT attack (Koh et al., 2018) and the min-max attack (Steinhardt et al., 2017; Koh et al., 2018). However, the KKT attack cannot scale well for multi-class classification and is limited to margin based losses (Koh et al., 2018). The min-max attack only works for indiscriminate attack setting, but additionally provides a certificate on worst case test loss for a fixed number of poisoning points. We are also inspired by Steinhardt et al. (2017) to adopt online convex optimization to instantiate our model-targeted attack, but now deals more general attack scenario. We also distinguish ourselves from the poisoning attack against online learning (Wang & Chaudhuri, 2018). The attack against online learning considers a setting where training data arrives in a streaming manner while we

consider the offline setting with training data being fixed. Another line of work studies "targeted" poisoning attacks where an adversary guarantees to increase the probability of an arbitrary "bad" property (Mahloujifar et al., 2019a; 2017; 2019b), as long as that property has some non-negligible chance of naturally happening. These attacks cannot be applied in the model-targeted setting as the probability of naturally producing a specific target model is often negligible. Related to our Theorem 4.2, Ma et al. (2019) also derived a lower bound on number of poisoning points (to induce a target model), but their lower bound only applies when differential privacy is deployed during the model training process (and hence hurts model utility), which is different from our problem setting.

## 4 POISONING ATTACK WITH A TARGET MODEL

Our new poisoning attack determines a target model and selects poisoning points to achieve that target model. The target model generation is not our focus and we adopt the heuristic approach proposed by Koh et al. (2018). For the new poisoning attack, first, we show the algorithm that generates the poisoning points in Section 4.1 and then prove that the generated poisoning points, once added to the clean data, can produce a classifier that asymptotically converges to the target classifier in Section 4.2.

### 4.1 MODEL-TARGETED POISONING WITH ONLINE LEARNING

The main idea of our model-targeted poisoning attack, as outlined in Algorithm 1, is to sequentially add a point into the training set that have maximum loss difference between the intermediate model obtained so far and the target model, and by training models on the updated training set, we actually minimize the gap in the loss of the intermediate classifier and the target classifier. Repeating the process then eventually generates classifiers that have similar loss distribution as the target classifier. We show in Section 4.2 why similar loss distribution implies convergence.

---

**Algorithm 1** ModelTargetedPoisoning

**Input:** $\mathcal{D}_c$, the loss functions ($L$ and $l$), $\theta_p$
**Output:** $\mathcal{D}_p$
  1: $\mathcal{D}_p = \emptyset$
  2: **while** stop criteria not met **do**
  3:      $\theta_t = \arg\min L(\theta; \mathcal{D}_c \cup \mathcal{D}_p)$
  4:      $(x^*, y^*) = \arg\max_{\mathcal{X} \times \mathcal{Y}} l(\theta_t; x, y) - l(\theta_p; x, y)$
  5:      $\mathcal{D}_p = \mathcal{D}_p \cup \{(x^*, y^*)\}$
  6: **end while**
  7: **return** $\mathcal{D}_p$

---

Algorithm 1 requires the input of clean training set $\mathcal{D}_c$, the Loss function ($L$ for set of points and $l$ for individual point) and the target model $\theta_p$. The output from Algorithm 1 will be the set of poisoning points $\mathcal{D}_p$. The algorithm is simple: first, adversaries train the intermediate model $\theta_t$ on the mixture of clean and poisoning points $\mathcal{D}_c \cup \mathcal{D}_p$ with $\mathcal{D}_p$ an empty set in first iteration (Line 3). The adversary then searches for the point that maximizes the loss difference between $\theta_t$ and $\theta_p$ (Line 4). After the point of maximum loss difference is found, it is added to the poisoning set $\mathcal{D}_p$ (Line 5). The whole process repeats until the stop condition is satisfied in Line 2. The stop condition is flexible and it can take various forms: 1) adversary has a budget $T$ on number of poisoning points, and the algorithm halts when the algorithm runs for $T$ iterations; 2) the intermediate classifier $\theta_t$ is closer to the target classifier (than a preset threshold $\epsilon$) in terms of the maximum loss difference, and more details regarding this distance metric will be introduced in Section 4.2; 3) adversary has some requirement on the accuracy and the algorithm terminates when $\theta_t$ satisfies the accuracy requirement. Since we focus on producing a classifier close to the target model, we adopt the second stop criterion that measures the distance with respect to the maximum loss difference, and report results based on this criterion in Section 5.

A nice property of Algorithm 1 is that the classifier $\theta_{atk}$ trained on $\mathcal{D}_c \cup \mathcal{D}_p$ is close to the target model $\theta_p$ and asymptotically converges to $\theta_p$. Details of the convergence will be shown in the next section. The algorithm may appear to be slow, particularly for larger models due to requirement of repeatedly training a model in line 3. However, this is not an issue. First, as will be shown in next section, the algorithm is an online optimization process and line 3 corresponds to solving the online optimization

problem exactly. However, people often use the very efficient online gradient descent method to approximately solve the problem and its asymptotic performance is the same (Shalev-Shwartz, 2012). Second, if we solve the optimization problem exactly, we can add multiple copies of $(x^*, y^*)$ into $\mathcal{D}_p$ each time. This reduces the overall iteration number, and hence reduces the number of times retraining models. The proof of convergence will be similar. For simplicity in interpreting the results, we do not use this in our experiments and add only one copy of $(x^*, y^*)$ each iteration. However, we also tested the performance by adding two copies of $(x^*, y^*)$ and find that the attack results are nearly the same while the efficiency is improved significantly. For example, for the experiments we tried on MNIST 1–7 dataset, by adding 2 copies of points, with same number of poisoning points, the attack success rate decreases at most by 0.7% while the execution time is reduced approximately by half.

## 4.2 CONVERGENCE OF OUR POISONING ATTACK

Before proving the convergence of Algorithm 1, we need to measure the distance of the model $\theta_{atk}$ trained on $\mathcal{D}_c \cup \mathcal{D}_p$ to the target model $\theta_p$. First, we define a general closeness measure based on their prediction performance which we will use to state our convergence theorem:

**Definition 1** (Loss-based distance and $\epsilon$-close). *For two models $\theta_1$ and $\theta_2$, a space $\mathcal{X} \times \mathcal{Y}$ and a loss function $l(\theta; x, y)$, we define loss-based distance $D_{l,\mathcal{X},\mathcal{Y}} \colon \Theta \times \Theta \to R$ as*

$$D_{l,\mathcal{X},\mathcal{Y}}(\theta_1, \theta_2) = \max_{(x,y) \in \mathcal{X} \times \mathcal{Y}} l(\theta_1; x, y) - l(\theta_2; x, y),$$

*and we say model $\theta_1$ is $\epsilon$-close to model $\theta_2$ when the loss-based distance from $\theta_1$ to $\theta_2$ is upper bounded by $\epsilon$.*

**Why is loss-based distance a meaningful notion of closeness?** We argue that this notion captures the "behavorial" distance between two models. Namely, if $\theta_1$ is $\epsilon$-close (as measured by loss-based distance) to $\theta_2$ and vice versa, then $\theta_1$ and $\theta_2$ would have almost equal loss on all the points, meaning that they have almost the same behavior across all the space. Note that our general definition of loss-based distance does not have the symmetry property of metrics and hence is not a metric. However, it has some other properties of metrics in the space of attainable models. For example, if some model $\theta$ is attainable using ERM, no model could have negative distance to it. To further show the value of this distance notion, in Appendix B we demonstrate an $O(\epsilon)$ upper bound on the $\ell_1$-norm of difference between two models that are $\epsilon$-close with respect to loss-based distance for the special case of Hinge loss. For Hinge loss, it also satisfies the *bi-directional closeness*, that is if $\theta_1$ is $\epsilon$-close to $\theta_2$, then $\theta_2$ is $O(\epsilon)$-close to $\theta$ (details can be found in Corollary B.2.1), and the proof details can be found in Appendix B. In the rest of the paper, we will use terms $\epsilon$-close or $\epsilon$-closeness to denote that a model is $\epsilon$ away from another model based on the loss-based distance.

Our convergence theorem uses the loss-based distance to establish that the attack of Algorithm 1 converges to the target classifier:

**Theorem 4.1.** *After at most $T$ steps, Algorithm 1 will produce the poisoning set $\mathcal{D}_p$ and the classifier trained on $\mathcal{D}_c \cup \mathcal{D}_p$ is $\epsilon$-close to $\theta_p$, with respect to loss-based distance, $D_{l,\mathcal{X},\mathcal{Y}}$, for*

$$\epsilon = \frac{\alpha(T) + L(\theta_p; D_c) - L(\theta_c; D_c)}{T \cdot \gamma}$$

*where, $\gamma$ is a constant for a given $\theta_p$ and classification task, and $\alpha(T)$ is the regret of the online algorithm when the loss function used for training is convex.*

**Remark 1.** *Online learning algorithms with sublinear regret bound can be applied to show the convergence. Here, we adopt results from McMahan (2017). Specifically, $\alpha(T)$ is in the order of $O(\log T)$ and we have $\epsilon \le O(\frac{\log T}{T})$ when the loss function is additionally Lipschitz continuous and the regularizer $R(\theta)$ is strongly convex, and $\epsilon \to 0$ when $T \to +\infty$. $\alpha(T)$ is also in the order of $O(\log T)$ when the loss function used for training is strongly convex and the regularizer is convex.*

**Proof idea.** The full proof of Theorem 4.1 is in Appendix A. Here, we only summarize the high level proof idea. The key idea is to frame the poisoning problem as an online learning problem. In this formulation, each step of the online learning problem corresponds to the $i$th poison point $(x_i, y_i)$. In particular, the loss function at iteration $i$ of the online learning problem is set to $l(\cdot; x_i, y_i)$. Then, we

show that by defining the parameters of the online learning problem in a careful way, the output of the follow-the-leader (FTL) algorithm (Shalev-Shwartz, 2012) at iteration $i$ is a model that is identical to training a model on a dataset consisting of the clean points and the first $i - 1$ poisoning points. On the other hand, the way the poisoning points are selected, we can show that at the $i$th iteration the maximum loss difference between the target model and the best induced model so far would be smaller than the regret of the FTL algorithm divided by the number of poisoning points. The convergence bound of Theorem 4.1 boils down to regret analysis of the algorithm based on the loss function. Since we are assuming the loss function is convex with a strongly convex regularizer (or a strongly convex loss function with a convex regularizer), we can show that the regret is bounded by $O(\log T)$ and hence the loss distance between the induced model and the target model converges to 0.

**Implications of Theorem 4.1** The theorem says that the loss-based distance of the model trained on $\mathcal{D}_c \cup \mathcal{D}_p$ to the target model correlates to the loss difference between the target model and the clean model $\theta_c$ (trained on $\mathcal{D}_c$) on $\mathcal{D}_c$, and correlates inversely with the number of poisoning points. Therefore, it implies 1) if the target classifier $\theta_p$ has lower loss on $\mathcal{D}_c$, then it is easier to achieve the target model, and 2) with more poisoning points, we get closer to the target classifier and our attack will be more effective. The theorem also justifies the motivation behind the heuristic method in Koh et al. (2018) to select a target classifier with lower loss on clean data. For the indiscriminate attack scenario, we also improve the heuristic approach by adaptively updating the model and producing target classifiers with much lower loss on the clean set. This helps to empirically validate our theorem. Details of the original and improved heuristic approach, and relevant experiments are in Appendix D.1.

### 4.3 Lower Bound on the Number of Poisoning Points

We first provide the lower bound on number of poisoning points required for producing the target classifier in addition only setting (Theorem 4.2), and then explain how the lower bound estimation can be incorporated into Algorithm 1. The intuition behind the theorem below is, when the number of poisoning points added to the clean training set is smaller than the lower bound, there always exists a classifier $\theta$ with lower loss compared to $\theta_p$ and hence the target classifier cannot be attained.

**Theorem 4.2** (Lower Bound)**.** *Given a target classifier $\theta_p$, to reproduce $\theta_p$ by adding the poisoning set $\mathcal{D}_p$ into $\mathcal{D}_c$, the number of poisoning points $|\mathcal{D}_p|$ cannot be lower than*

$$\sup_\theta z(\theta) = \frac{L(\theta_p; \mathcal{D}_c) - L(\theta; \mathcal{D}_c) + NC_R(R(\theta_p) - R(\theta))}{\sup_{x,y}\left(l(\theta; x, y) - l(\theta_p; x, y)\right) + C_R(R(\theta) - R(\theta_p))}.$$

**Corollary 4.2.1.** *If we further assume bi-directional closeness in the loss-based distance, we can also derive the lower bound on number of poisoning points needed to induce models that are $\epsilon$-close to the target model. More precisely, if $\theta_1$ being $\epsilon$-close to $\theta_2$ implies that $\theta_2$ is also $k \cdot \epsilon$ close to $\theta_1$, then we have,*

$$\sup_\theta z^{'}(\theta) = \frac{L(\theta_p; \mathcal{D}_c) - L(\theta; \mathcal{D}_c) - NC_R \cdot R^* - Nk\epsilon}{\sup_{x,y}\left(l(\theta; x, y) - l(\theta_p; x, y)\right) + C_R \cdot R^* + k\epsilon}.$$

*where $R^*$ is an upper bound on the nonnegative regularizer $R(\theta)$.*

The formula for the lower bound in Theorem 4.2 (and also the lower bound in Corollary 4.2.1) can be easily incorporated into Algorithm 1 to obtain tighter theoretical lower bound. We simply need to check all of the intermediate classifier $\theta_t$ produced during the attack process and replace $\theta$ with $\theta_t$, and the lower bound can be computed for the pair of $\theta_t$ and $\theta_p$. Algorithm 1 then additionally returns the lower bound, which is the highest lower bound computed from our poisoning procedure.

## 5 Experiments

We first describe our experimental setup regarding the datasets, models, attacks and target classifiers. Next, we present the experimental results by showing the convergence of Algorithm 1, the comparison of attack success rates to state-of-the-art poisoning attack, and the theoretical lower bound for inducing a given target classifier and its gap to the number of poisoning points used by our attack.

We are most interested in subpopulation attacks, since they correspond to the more realistic attacker goal of impacting the classifier outputs for a targeted subpopulation. Therefore, in the main body, we introduce the results of SVM model on the Adult dataset (Dua & Graff, 2017) in the subpopulation poisoning scenario. For completeness, we also evaluate our attack on SVM model on MNIST 1–7 dataset in the indiscriminate poisoning scenario but defer details on those experiments to Appendix D. Our findings for the indiscriminate attacks are that the attack gradually and consistently converges to the target model in terms of the maximum loss difference and the Euclidean distance to the target, with attack success rates that are comparable to the state-of-the-art attack (unlike the subpopulation attacks, where our attack produces superior results). To further verify the universal effectiveness of our attack, we also evaluate our attack on additional dataset (Dogfish) and model (logistic regression), and more details can be found in Appendix F.

**Dataset, Model and Attacks.** For the subpopulation attack experiments, we use the Adult dataset (Dua & Graff, 2017). This dataset was used for evaluation by the first subpopulation attack paper (Jagielski et al., 2019). We downsampled the Adult dataset to ensure it is class-balanced and we ended up having 15,682 training and 7,692 test examples. We conduct experiments on linear SVM model and compare our model-targeted poisoning attack in Algorithm 1 to the state-of-the-art KKT attack (Koh et al., 2018). We do not include the model-targeted attack from Mei & Zhu (2015b) because it underperforms the KKT attack (Koh et al., 2018). We also do not include objective-driven attacks because our main goal here is to evaluate how well our attack approaches a given target model, across a range of target models. Model-targeted attacks can be compared to objective-driven attacks with regards to a given attacker objective by choosing the target model in a careful way. We show some heuristics of choosing such target models and comparison to some objective-driven attacks in Appendix E.

Both our attack and the KKT attack take as input a target classifier and the original training data, and output a set of poisoning points selected with the goal that the induced classifier is as close as possible to the target classifier. We compare the effectiveness of the attacks in selecting poisoning points that converge to a given target classifier by testing the attacks using the same target model.

The KKT attack requires a target number of poisoning points as an input while our attack is more flexible and can either take a target number of poisoning points or a threshold for $\epsilon$-close distance to the target model. Since we do not know the number of poisoning points needed to reach some attacker goal in advance for the KKT attack, we first run our attack and produce a classifier that satisfies the selected $\epsilon$-close distance threshold. The loss function is set as the hinge loss since we target an SVM model in our experiments and we set $\epsilon = 0.01$ for all these experiments. Then, we use the size of the poisoning set returned from our attack (denoted by $n_p$) as the input to the KKT attack for the target number of poisons needed. We also compare the two attacks with varying numbers of poisoning points up to $n_p$. For the KKT attack, its entire optimization process must be rerun whenever the target number of poisoning points changes. Hence, it is infeasible to evaluate the KKT attack on many different poisoning set sizes. In our experiments, we run the KKT attack five poisoning set sizes: $0.2 \cdot n_p$, $0.4 \cdot n_p$, $0.6 \cdot n_p$, $0.8 \cdot n_p$, and $n_p$. In contrast, we simply run our attack for iterations up to the maximum number of poisoning points, collecting a data point for iteration up to $n_p$.

**Subpopulations.** We identify the subpopulations for the Adult dataset using $k$-means clustering techniques (ClusterMatch (Jagielski et al., 2019)) to obtain different clusters ($k = 20$ in our case). For each cluster, we select instances with label "<=50K" to form the subpopulation (indicating all instances in the subpopulation are in low income group). This way of defining subpopulation is rather arbitrary (in constrast to a more likely attack goal which would select subpopulations based on demographic characteristics), but enables us to simplify the analysis. From the 20 subpopulations obtained, we select three subpopulations with the highest test accuracy on the clean model and they all have 100% test accuracy, indicating all instances in these subpopulations are correctly classified as low income. This enables us to use "attack success rate" and "accuracy" without any ambiguity on the subpopulation—for each of our subpopulations, all instances are originally classified as low income, and the simulated attacker's goal is to have them classified as high income.

For each subpopulation, we use the heuristic approach from Koh et al. (2018) to generate a target classifier that has 0% accuracy (100% attacker success) on the subpopulation, indicating that all subpopulation instances are now classified as high income.

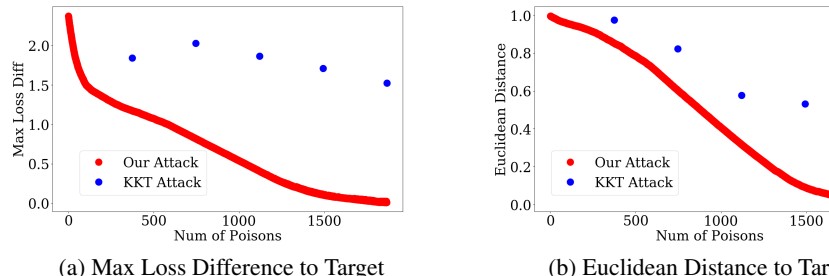

(a) Max Loss Difference to Target                    (b) Euclidean Distance to Target

Figure 1: Attack convergence (results shown are for the first subpopulation, Cluster 0). The maximum number of poisons is set using the 0.01-close threshold to target classifier.

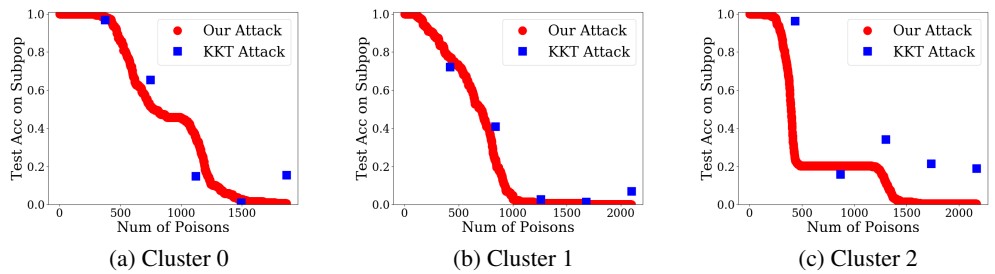

(a) Cluster 0                    (b) Cluster 1                    (c) Cluster 2

Figure 2: Test accuracy for each subpopulation with classifiers induced by poisoning points obtained from our attack and the KKT attack.

|                   | Cluster 0        | Cluster 1        | Cluster 2        |
| ----------------- | ---------------- | ---------------- | ---------------- |
| Poisoning Points  | 1866             | 2097             | $2163.3 \pm 2.5$ |
| Lower Bound       | $1666.7 \pm 2.4$ | $1831.4 \pm 5.0$ | $1863.0 \pm 9.2$ |

Table 1: Poisoning points needed to achieve target classifiers induced from our attack. Top row means number of poisoning points used by our attack. Bottom row means the lower bound computed from Theorem 4.2 for the induced classifiers. All results are averaged over 4 runs, integer value in the cell means we get exactly same value for 4 runs and others are shown with the average and standard error.

**Convergence.** Figure 1 shows the convergence of Algorithm 1 using both maximum loss difference and Euclidean distance to the target. The maximum number of poisons ($n_p$) for the experiments is obtained when the classifier from Algorithm 1 is 0.01-close to the target classifier. Our attack steadily reduces the maximum loss difference and Euclidean distance to the target model, in contrast to the KKT attack which does not seem to converge towards the target model reliably. Concretely, at the maximum number of poisons in Figure 1, both the maximum loss difference and Euclidean distance of our attack (to the target) is less than 2% of the corresponding distances of the KKT attack.

**Attack Success.** Next, we compare the classifiers induced by the two attacks in terms of the attacker's goal of reducing the test accuracy on the subpopulation. Figure 2 shows the accuracy results for the three subpopulations. For each test, the maximum number of poisoning points is obtained by running our attack with a target of 0.01-closeness (in loss-based distance). For the three subpopulations, at the maximum number of poisons, our attack is much more successful than the KKT attack—the induced classifiers have 0.5% accuracy compared to 15.4% accuracy for KKT on subpopulation 1, 0.0% compared to 6.9% on subpopulation 2, and 0.3% compared to 20.1% on subpopulation 3.

**Near Optimality of Our Attack.** In order to show the optimality of our attack, we calculate a lower bound on the number of poisoning points needed to induce the model that is induced by the poisoning points that are found by our attack. We calculate this lower bound on the number of poisons using Theorem 4.2 (details in Section 4.3). Note that Theorem 4.2 provides a valid lower bound based on any intermediate model. In order to get a lower bound on the number of poisoning points, we

only need to use Theorem 4.2 on the encountered intermediate models and report the best one. We do this by running Algorithm 1 using the induced model (and not the previous target model) as the target model, terminating when the induced classifier is 0.01-close to the given target model. We then consider all the intermediate classifiers that the algorithm induced across the iterations. Our calculated lower bound in Table 1 shows that the gap between the lower bound and the number of used poison points is relatively small. This means our attack is nearly optimal in terms of minimizing the number of poisoning points needed to induce the target classifier.

## 6 CONCLUSION AND DISCUSSION

We propose a general poisoning framework with provable guarantees to reach any achievable target classifier, along with a lower bound on the number of poisoning points needed. Our attack is a generic tool that first captures the goal of adversary as a target model, and then focuses on the power of attacks to induce that model. This separation enables future work to explore the effectiveness of poisoning attacks corresponding to different adversarial goals. Our framework also applies in scenarios where adversaries first remove points and then add new points into the training set. We have not considered defenses in this work, and it is important to study the effectiveness of our attack against data poisoning defenses. Defenses may be designed the limit the search space of the points with maximum loss difference and increasing the number of poisoning points needed.

One limitation of our framework is the requirement in the concavity of the difference of loss functions to efficiently search for its maximum value. However, our approach might still be effective in these cases by using local optimization techniques to search for poisoning points with (approximate) maximum loss difference and we have demonstrated this in the case of logistic loss. More formally, if the approximate maximum loss difference $\hat{l}$ found from local optimization techniques is within a constant factor from the globally optimal value $l^*$ (i.e., $\hat{l} \geq \alpha l^*, 0 < \alpha < 1$), then we still enjoy similar convergence guarantees. It is important to note that the convergence property of our attack holds (with strongly convex regularizer) for any Lipschitz and convex loss function, and does not require the loss difference to be concave. The theoretical guarantees in the paper do not apply to non-convex models, although it might be possible to empirically apply our attack to these models. Incorporating online learning for non-convex functions might be one possible path to extend our theoretical analysis into non-convex settings.

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

## A  PROOFS

In this section, we provide the proofs of the main theorems shown in this paper. For convenience, we restate all the theorems below while also referencing to the main paper.

Before proving the main theorem, we introduce two new definitions and several lemmas to assist with the proof.

**Definition 2** (Attainable models). *We say $\theta$ is $C_R$-attainable with respect to loss function $l$ and regularization function $R$ if there exists a training set $\mathcal{D}$ such that*

$$\theta = \arg\min_{\theta \in \Theta} \frac{1}{|D|} \cdot L(\theta; \mathcal{D}) + C_R \cdot R(\theta)$$

**Lemma A.1.** *Let $\theta_1$ and $\theta_2$ be two $C_R$-attainable parameters for some $C_R > 0$ such that $R(\theta_1) > R(\theta_2)$. Then,*

$$\sup_{x,y} \big(l(\theta_2; x, y) - l(\theta_1; x, y)\big) / \big(R(\theta_1) - R(\theta_2)\big) > C_R.$$

*Proof.* Consider any attainable pairs of $(\theta_1, \theta_2)$ such that $R(\theta_1) > R(\theta_2)$ and let $\mathcal{D}_1$ to be training set that the training algorithm produces the unique minimizer $\theta_1$. Namely,

$$\theta_1 = \arg\min_{\theta} \frac{1}{|\mathcal{D}_1|} \cdot L(\theta; \mathcal{D}_1) + C_R \cdot R(\theta)$$

Since $\theta_1$ minimizes the total loss on $\mathcal{D}_1$ uniquely, we have

$$\frac{1}{|\mathcal{D}_1|} L(\theta_2; \mathcal{D}_1) + C_R \cdot R(\theta_2) > \frac{1}{|\mathcal{D}_1|} L(\theta_1; \mathcal{D}_1) + C_R \cdot R(\theta_1)$$

By rearranging the above inequality and by an averaging argument, we have

$$\sup_{x,y} \big(l(\theta_2; x, y) - l(\theta_1; x, y)\big) \geq \frac{1}{|\mathcal{D}_1|} L(\theta_2; \mathcal{D}_1) - \frac{1}{|\mathcal{D}_1|} L(\theta_1; \mathcal{D}_1) > C_R \cdot \big(R(\theta_1) - R(\theta_2)\big).$$

Now since $R(\theta_1) > R(\theta_2)$ we have

$$\sup_{x,y} \big(l(\theta_2; x, y) - l(\theta_1; x, y)\big) / \big(R(\theta_1) - R(\theta_2)\big) > C_R.$$

$\square$

**Lemma A.2.** *Let $F$ be the family of all $C_R$-attainable models. For any $\theta_1 \in F$, there is a constant $\gamma$ where for all $\theta_2 \in F$ we have*

$$\sup_{x,y} \big(l(\theta_2; x, y) - l(\theta_1; x, y)\big) + C_R(R(\theta_2) - R(\theta_1)) > \gamma \cdot \sup_{x,y} \big(l(\theta_2; x, y) - l(\theta_1; x, y)\big)$$

*where $\gamma$ is a positive constant related to, $\theta_1$, $C_R$ and other model parameters (fixed for a given classification task).*

*Proof.* We prove the lemma for $\gamma = 1 - C_R/C$ for

$$C = \left( \inf_{\substack{\theta_2 \in F \\ \text{s.t. } R(\theta_1) > R(\theta_2)}} \sup_{x,y} (l(\theta_2; x, y) - l(\theta_1; x, y)) / (R(\theta_1) - R(\theta_2)) \right).$$

First, note that by Lemma A.1 we have

$$C > C_R \geq 0. \tag{2}$$

which implies $\gamma$ is positive. Now we consider two subcases based on the sign of $R(\theta_2) - R(\theta_1)$:

**Case 1:** $R(\theta_2) - R(\theta_1) \geq 0$. In this case the inequality is straightforward:

$$\sup_{x,y} \big(l(\theta_2; x, y) - l(\theta_1; x, y)\big) + C_R \cdot (R(\theta_2) - R(\theta_1)) \geq \sup_{x,y} \big(l(\theta_2; x, y) - l(\theta_1; x, y)\big)$$
$$> (1 - C_R/C) \cdot \sup_{x,y} \big(l(\theta_2; x, y) - l(\theta_1; x, y)\big),$$

where the last inequality is based on equation 2.

**Case 2:** $R(\theta_2) - R(\theta_1) < 0$. From the definition of $C$ we have

$$R(\theta_1) - R(\theta_2) \leq \frac{\sup_{x,y} \big(l(\theta_2; x, y) - l(\theta_1; x, y)\big)}{C}.$$

Equivalently, we can say

$$R(\theta_2) - R(\theta_1) \geq -\frac{\sup_{x,y} \big(l(\theta_2; x, y) - l(\theta_1; x, y)\big)}{C}.$$

Replacing $R(\theta_2) - R(\theta_1)$ with the lower bound above completes the proof, namely

$$\sup_{x,y} \big(l(\theta_2; x, y) - l(\theta_1; x, y)\big) + C_R(R(\theta_2) - R(\theta_1)) \geq (1 - C_R/C) \cdot \sup_{x,y} \big(l(\theta_2; x, y) - l(\theta_1; x, y)\big).$$

$\square$

With Definition 2 and the lemmas, we are ready to prove Theorem 4.1 (restating Theorem 4.1, from Section 4.2):

**Theorem 4.1.** *After at most $T$ steps, Algorithm 1 will produce the poisoning set $\mathcal{D}_p$ and the classifier trained on $\mathcal{D}_c \cup \mathcal{D}_p$ is $\epsilon$-close to $\theta_p$, with respect to loss-based distance, $D_{l,\mathcal{X},\mathcal{Y}}$, for*

$$\epsilon = \frac{\alpha(T) + L(\theta_p; D_c) - L(\theta_c; D_c)}{T \cdot \gamma}$$

*where, $\gamma$ is a constant for a given $\theta_p$ and classification task, and $\alpha(T)$ is the regret of the online algorithm when the loss function used for training is convex.*

The goal of the adversary is to get $\epsilon$-close to $\theta_p$ (in terms of the loss-based distance) by injecting (potentially few) number of poisoned training data. The algorithm is in essence an online learning problem and we transform Algorithm 1 into the form of standard online learning problem. Specifically, we adopt the *follow the leader* (FTL) framework to describe Algorithm 1 in the language of standard online learning problem. We first describe the online learning setting considered in this paper and the notion of the regret.

**Definition 3.** *Let $\mathcal{L}$ be a class of loss functions, $\Theta$ set of possible models, $A: (\Theta \times \mathcal{L})^* \to \Theta$ an online learner and $S: (\Theta \times \mathcal{L})^* \times \Theta \to \mathcal{L}$ a strategy for picking loss functions in different rounds of online learning (adversarial environment in the context of online convex optimization). We use $\mathsf{Regret}(A, S, T)$ to denote the regret of $A$ against $S$, in $T$ rounds. Namely,*

$$\mathsf{Regret}(A, S, T) = \sum_{j=0}^{T} l_j(\theta_j) - \min_{\theta \in \Theta} \sum_{j=0}^{T} l_j(\theta)$$

*where*

$$\theta_i = A\big((\theta_0, l_0), \dots, (\theta_{i-1}, l_{i-1})\big) \quad and \quad l_i = S\big((\theta_0, l_0), \dots, (\theta_{i-1}, l_{i-1}), \theta_i\big).$$

With the online learning problem set up, we proceed to the main proof which first describes Algorithm 1 in the FTL framework.

*Proof of Theorem 4.1.* The FTL framework proceeds by solving all the functions incurred during the previous online optimization steps, namely, $A_{\mathsf{FTL}}((\theta_0, l_0), \dots, (\theta_i, l_i)) = \arg\min_{\theta \in \Theta} \sum_{j=0}^{i} l_i(\theta)$.

Next, we describe how we design the $i$th loss function $l_i$ in each round of the online optimization. For the first choice, $A_{\mathsf{FTL}}$ chooses a random model $\theta_0 \in \Theta$. In the first round (round 0), $S_{\theta_p}$ uses the clean training set $\mathcal{D}_c$ and the loss is set as

$$S_{\theta_p}(\theta_0) = l_0(\theta) = L(\theta; \mathcal{D}_c) + N \cdot C_R \cdot R(\theta).$$

According to the FTL framework, $A_{\mathsf{FTL}}$ returns model that minimizes the loss on the clean training set $\mathcal{D}_c$ using the structural empirical risk minimization. For the subsequent iterations ($i \geq 1$), the loss functions is defined as, given the latest model $\theta_i$, $S_{\theta_p}$ first finds $(x_i^*, y_i^*)$ that maximizes the loss difference between $\theta_i$ and a target model $\theta_p$. Namely,

$$(x_i^*, y_i^*) = \underset{(x,y)}{\arg\max}\, l(\theta_i; x, y) - l(\theta_p; x, y)$$

and then chooses the $i$th loss function as follows:

$$S_{\theta_p}\big((\theta_0, l_0), \ldots, (\theta_{i-1}, l_{i-1}), \theta_i\big) = l_i(\theta) = l(\theta; x_i^*, y_i^*) + C_R \cdot R(\theta).$$

Now we will see how FTL framework behaves when working on these loss functions at different iterations. We use $D_p^i$ to denote the set $\{(x_1^*, y_1^*), \ldots, (x_i^*, y_i^*)\}$. We have

$$
\begin{aligned}
\theta_i = A_{\mathsf{FTL}}((\theta_0, l_0), \ldots, (\theta_{i-1}, l_{i-1})) &= \underset{\theta \in \Theta}{\arg\min} \sum_{j=0}^{i-1} l_j(\theta) \\
&= \underset{\theta \in \Theta}{\arg\min}\, L(\theta; \mathcal{D}_c) + N \cdot C_R \cdot R(\theta) \\
&\quad + \sum_{j=1}^{i-1} l(\theta; x_i^*, y_i^*) + C_R \cdot R(\theta) \\
&= \underset{\theta \in \Theta}{\arg\min}\, L(\theta; \mathcal{D}_c \cup \mathcal{D}_p^{i-1}) + (N + i - 1) \cdot C_R \cdot R(\theta) \\
&= \underset{\theta \in \Theta}{\arg\min}\, \frac{1}{|\mathcal{D}_c \cup \mathcal{D}_p^{i-1}|} L(\theta; \mathcal{D}_c \cup \mathcal{D}_p^{i-1}) + C_R \cdot R(\theta)
\end{aligned}
$$

This means that $A_{\mathsf{FTL}}$ algorithm, at each step, trains a new model over the combination of clean data and poison data so far ($i - 1$ number of poisons). Now we want to see what is the translation of the $\mathrm{Regret}(A_{\mathsf{FTL}}, S_{\theta_p}, T)$. If we can prove an upper bound on regret, namely if we show $\mathrm{Regret}(A_{\mathsf{FTL}}, S_{\theta_p}, T) \leq \alpha(T)$ for some function $\alpha$, then we have

$$\sum_{j=0}^{T} l_j(\theta_j) - \sum_{j=0}^{T} l_j(\theta_p) \leq \sum_{j=0}^{T} l_j(\theta_j) - \underset{\theta \in \Theta}{\min} \sum_{j=0}^{T} l_j(\theta) \leq \alpha(T)$$

which implies

$$
\begin{aligned}
\sum_{j=0}^{T} l_j(\theta_j) - \sum_{j=0}^{T} l_j(\theta_p) &= L(\theta_c; D_c) - L(\theta_p; D_c) + N \cdot C_R \cdot (R(\theta_c) - R(\theta_p)) \\
&\quad + \sum_{j=1}^{T} l_j(\theta_j) - \sum_{j=1}^{T} l_j(\theta_p) \\
&= L(\theta_c; D_c) - L(\theta_p; D_c) + N \cdot C_R \cdot (R(\theta_c) - R(\theta_p)) \\
&\quad + \sum_{j=1}^{T} \Big[ \underset{x,y}{\max}\, \big(l(\theta_j; x, y) - l(\theta_p; x, y)\big) + C_R \cdot (R(\theta_j) - R(\theta_p)) \Big] \\
&\leq \alpha(T)
\end{aligned}
$$

Therefore we have

$$\sum_{j=1}^{T} \left[ \max_{x,y} \left( l(\theta_j; x, y) - l(\theta_p; x, y) \right) + C_R \cdot (R(\theta_j) - R(\theta_p)) \right] \leq \alpha(T) + L(\theta_p; D_c) - L(\theta_c; D_c)$$
$$+ N \cdot C_R \cdot (R(\theta_p) - R(\theta_c))$$

Based on Lemma A.2, we further have

$$\sum_{j=1}^{T} \gamma \cdot \left( \max_{x,y} l(\theta_j; x, y) - l(\theta_p; x, y) \right) \leq \alpha(T) + L(\theta_p; D_c) - L(\theta_c; D_c)$$
$$+ N \cdot C_R \cdot (R(\theta_p) - R(\theta_c))$$

Above inequality states that average of the maximum loss difference in all previous rounds is bounded from above. Therefore, we know that among the $T$ iterations, there exist an iteration $j^* \in [T]$ (with lowest maximum loss difference) such that the maximum loss difference of $\theta_{j^*}$ is $\epsilon$-close to $\theta_p$ with respect to the loss-based distance where

$$\epsilon = \frac{\alpha(T) + L(\theta_p; D_c) - L(\theta_c; D_c) + N \cdot C_R \cdot (R(\theta_p) - R(\theta_c))}{T \cdot \gamma}.$$

□

Theorem 4.1 characterizes the dependencies of $\epsilon$ on $\alpha(T)$ and the constant term $L(\theta_p; D_c) - L(\theta_c; D_c) + N \cdot C_R \cdot (R(\theta_p) - R(\theta_c))$. To show the convergence of Algorithm 1, we need to ensure $\epsilon \to 0$ when $T \to +\infty$, which implies we need to show $\alpha(T) \leq O(\sqrt{T})$. Following remark (restating Remark 1 in Section 4.2) and its proof shows the desired convergence.

**Remark 1.** *Online learning algorithms with sublinear regret bound can be applied to show the convergence. Here, we adopt the regret analysis from McMahan (2017). Specifically, $\alpha(T)$ is in the order of $O(\log T)$) and we have $\epsilon \leq O(\frac{\log T}{T})$ when the loss function is Lipschitz continuous and the regularizer $R(\theta)$ is strongly convex, and $\epsilon \to 0$ when $T \to +\infty$. $\alpha(T)$ is also in the order of $O(\log T)$ when the loss function used for training is strongly convex and the regularizer is convex.*

Our FTL framework formulation can utilize the existing logarithmic regret bound of adaptive FTL algorithm when the objective functions are strongly convex with respect to some norm $\| \cdot \|$, as illustrated in Section 3.6 in McMahan (2017). For clarity in presentation, we first restate their related results below.

**Setting 1** (Setting 1 in McMahan (2017)). *Given a sequence of objective loss functions $f_1, f_2, ..., f_i$ and a sequence of incremental regularization functions $r_0, r_1, ..., r_i$ we consider an algorithm that selects the response point based on*

$$\theta_1 = \arg\min_{\theta \in \mathbb{R}^d} r_0(\theta)$$

$$\theta_{i+1} = \arg\min_{\theta \in \mathbb{R}^d} \sum_{j=1}^{i} f_j(\theta) + r_j(\theta) + r_0(\theta), \text{for } i = 1, 2, ...$$

*We simplify the summation notation with $f_{1:i}(\theta) = \sum_{j=1}^{i} f_j(\theta)$. Assume that $r_i$ is a convex function and satisfy $r_i(\theta) \geq 0$ for $i \in \{0, 1, 2, ...\}$, against a sequence of convex loss functions $f_i : \mathbb{R}^d \to R \cup \{\infty\}$. Further, letting $h_{0:i} = r_{0:i} + f_{1:i}$ we assume $\text{dom } h_{0:i}$ is non-empty. Recalling $\theta_i = \arg\min_\theta h_{0:i-1}(\theta)$, we further assume $\partial f_i(\theta_i)$ is non-empty. We denote the dual norm of a norm $\| \cdot \|$ as $\| \cdot \|_*$.*

**Theorem A.3** (Restatement of Theorem 1 in McMahan (2017)). *Consider Setting 1, and suppose the $r_i$ are chosen such that $r_{0:i} + f_{1:i+1}$ is 1-strongly-convex w.r.t. some norm $\| \cdot \|_{(i)}$.. If we define the regret of the algorithm with respect to a selected point $\theta^*$ as*

$$\text{Regret}_T(\theta^*, f_i) \equiv \sum_{i=1}^{T} f_i(\theta_i) - \sum_{i=1}^{T} f_i(\theta^*).$$

*Then, for any $\theta^* \in \mathbb{R}^d$ and for any $T > 0$, with $g_i \in \partial f_i(\theta_i)$, we have*

$$\text{Regret}_T(\theta^*, f_i) \leq r_{0:T-1}(\theta^*) + \frac{1}{2}\|g_i\|_{(i-1),*}^2$$

**Corollary A.3.1** (Formalization of FTL result in Section 3.6 in McMahan (2017)). *In the FTL framework (no individual regularizer is used in the optimization procedure), suppose each loss function $f_i$ is 1-strongly convex w.r.t. a norm $\|\cdot\|$, then we have*

$$\text{Regret}_T(\theta^*, f_i) \leq \frac{1}{2}\sum_{i=1}^T \frac{1}{i}\|g_i\|_*^2 \leq \frac{G^2}{2}(1 + \log T)$$

*with $\|g_i\|_* \leq G$.*

*Proof. The following proof is a restatement of the proof in Section 3.6 in McMahan (2017).* The proof follows from Theorem A.3. Since we are considering the FTL framework, let $r_i(\theta) = 0$ for all $i$ and define $\|\theta\|_{(i)} = \sqrt{i}\|\theta\|$. Observe that $h_{0:i}$ (i.e., $f_{1:i}$) is 1-strongly convex with respect to $\|\theta\|_{(i)}$ (Lemma 3 in McMahan (2017)), and we have $\|\theta\|_{(i),*} = \frac{1}{\sqrt{i}}\|\theta\|_*$. Then by applying Theorem A.3, we have

$$\text{Regret}_T(\theta^*, f_i) \leq \frac{1}{2}\sum_{i=1}^T \|g_i\|_{(i),*}^2 = \frac{1}{2}\sum_{i=1}^T \frac{1}{i}\|g_i\|_*^2$$

Based on the inequality of $\sum_{i=1}^T 1/i \leq 1 + \log T$ and if we further assume $\|g_i\|_* \leq G$, then we can have

$$\frac{1}{2}\sum_{i=1}^T \frac{1}{i}\|g_i\|_*^2 \leq \frac{G^2}{2}(1 + \log T)$$

$\square$

*Proof of Remark 1.* We will prove the logarithmic regret bound in Remark 1 utilizing Corollary A.3.1. First of all, our online learning process fits into Setting 1. Specifically, we set $r_i(\theta) = 0$ for all $i$. For $f_i(\theta)$, when $1 \leq i \leq N$, we set $f_i(\theta) = \frac{1}{N}L(\theta; \mathcal{D}_c) + C_R \cdot R(\theta)$ (evenly distributing the term $L(\theta; \mathcal{D}_c) + N \cdot C_R \cdot R(\theta)$ across $N$ iterations) and when $i \geq N+1$, we set $f_i(\theta) = l_{i-N}(\theta)$. Details of $l_i$ can be referred from the proof of Theorem 4.1. Therefore, $f_i$ is 1-strongly convex with respect to a norm $\|\cdot\|$ (the norm is determined by the regularizer $R(\theta)$ and $C_R$). Further, $l_{0:i}(\theta) = f_{1:N+i}(\theta)$. In addition, the assumption that dom $h_{0:i}$ is non-empty in Setting 1 means when if we train a classifier on the poisoned data set, we can always return a model and hence the assumption is satisfied. The assumption of the existence of subgradient $\partial f_i(\theta_i)$ in Setting 1 is also satisfied by the poisoning attack scenario.

The logarithmic regret of $\text{Regret}(A_{\text{FTL}}, S_{\theta_p}, T)$ of our algorithm then follows from the result of $\text{Regret}_T(\theta^*, f_i)$ in Corollary A.3.1. Specifically, $l_{0:i}(\theta) = f_{1:N+i}(\theta)$ is 1-strongly convex to norm $\|\cdot\|_i = \sqrt{N+i}\|\cdot\|$ and since we assume the loss function is $G$-Lipschitz, we have $\|g_i\|_* \leq G$. Therefore, we have the logarithmic regret bound as:

$$\text{Regret}(A_{\text{FTL}}, S_{\theta_p}, T) \leq \alpha(T) = \frac{1}{2}\sum_{i=1}^T \frac{1}{i+N}\|g_i\|_*^2 \leq \frac{1}{2}\sum_{i=1}^T \frac{1}{i}\|g_i\|_*^2 \leq \frac{G^2}{2}(1+\log T) \leq O(\log T).$$

$\square$

We next provide the proof of the certified lower bound (restating Theorem 4.2 from Section 4.3):

**Theorem 4.2.** *Given a target classifier $\theta_p$, to reproduce $\theta_p$ by adding the poisoning set $\mathcal{D}_p$ into $\mathcal{D}_c$, the number of poisoning points $|\mathcal{D}_p|$ cannot be lower than*

$$\sup_{\theta} z(\theta) = \frac{L(\theta_p; \mathcal{D}_c) - L(\theta; \mathcal{D}_c) + NC_R(R(\theta_p) - R(\theta))}{\sup_{x,y} \left(l(\theta; x, y) - l(\theta_p; x, y)\right) + C_R(R(\theta) - R(\theta_p))}.$$

The main intuition behind the theorem is, when the the number of poisoning points added to the clean training set is lower than the certified lower bound, for structural empirical risk minimization problem (shown in equation 1 in the main paper), then target classifier will always have higher loss than another classifier and hence cannot be achieved.

*Proof.* We first show that for all models $\theta$, we can derive a lower bound on the number of poison points required to get $\theta_p$. Then since these lower bounds all hold, we can take the maximum over all of them and get a valid lower bound. We first show that for any model $\theta$, the minimum number of poisoning points cannot be lower than

$$z(\theta) = \frac{L(\theta_p; \mathcal{D}_c) - L(\theta; \mathcal{D}_c) + NC_R(R(\theta_p) - R(\theta))}{\sup_{x,y} \left(l(\theta; x, y) - l(\theta_p; x, y)\right) + C_R(R(\theta) - R(\theta_p))}.$$

Let us denote the point corresponding to the supremum of the loss difference between $\theta$ and $\theta_p$ as $(x^*, y^*)$ [2]. Namely, $l(\theta; x^*, y^*) - l(\theta_p; x^*, y^*) = \sup_{x,y} \left(l(\theta; x, y) - l(\theta_p; x, y)\right)$. Now suppose we can obtain $\theta_p$ with lower number of poisoning points $\underline{z} < z(\theta)$. Assume there is a poisoning set $\mathcal{D}_p$ with size $\underline{z}$ such that when added to $\mathcal{D}_c$ would result in $\theta_p$. We have

$$\sup_{x,y} \left(l(\theta; x, y) - l(\theta_p; x, y)\right) \geq \frac{1}{|\mathcal{D}_c \cup \mathcal{D}_p|} L(\theta; \mathcal{D}_c \cup \mathcal{D}_p) - \frac{1}{|\mathcal{D}_c \cup \mathcal{D}_p|} L(\theta_p; \mathcal{D}_c \cup \mathcal{D}_p)$$
$$> C_R \cdot \left(R(\theta_p) - R(\theta)\right),$$

implying $\sup_{x,y} \left(l(\theta; x, y) - l(\theta_p; x, y)\right) + C_R \cdot (R(\theta) - R(\theta_p)) > 0$. Based on the assumption that $\underline{z} < z(\theta)$, and the fact that $\sup_{x,y} \left(l(\theta; x, y) - l(\theta_p; x, y)\right) + C_R \cdot (R(\theta) - R(\theta_p)) > 0$, we have

$$\underline{z} \cdot \left(l(\theta; x^*, y^*) - l(\theta_p; x^*, y^*) + C_R(R(\theta) - R(\theta_p))\right)$$
$$< z(\theta) \cdot \left(l(\theta; x^*, y^*) - l(\theta_p; x^*, y^*) + C_R(R(\theta) - R(\theta_p))\right)$$
$$= L(\theta_p; \mathcal{D}_c) - L(\theta; \mathcal{D}_c) + NC_R(R(\theta_p) - R(\theta)).$$

where the equality is based on the definition of $z(\theta)$. On the other hand, by definition of $(x^*, y^*)$ for any $D_p$ of size $\underline{z}$, we have

$$L(\theta; D_p) - L(\theta_p, D_p) + \underline{z} \cdot (C_R \cdot R(\theta) - C_R \cdot R(\theta_p))$$
$$\leq \underline{z} \cdot \left(l(\theta; x^*, y^*) - l(\theta_p; x^*, y^*) + C_R(R(\theta) - R(\theta_p))\right).$$

The above two inequalities imply that for any set $D_p$ with size $\underline{z}$ we have

$$\frac{1}{|\mathcal{D}_c \cup \mathcal{D}_p|} L(\theta; \mathcal{D}_c \cup \mathcal{D}_p) + C_R \cdot R(\theta) < \frac{1}{|\mathcal{D}_c \cup \mathcal{D}_p|} L(\theta_p; \mathcal{D}_c \cup \mathcal{D}_p) + C_R \cdot R(\theta_p).$$

which indicates that adding $\mathcal{D}_p$ poisoning points into the training set $\mathcal{D}_c$, the model $\theta$ has lower loss compared to $\theta_p$, which is a contradiction to the assumption that $\theta_p$ has lowest loss on $\mathcal{D}_c \cup \mathcal{D}_p$ and can be achieved. Now, since $\theta_p$ needs to have lower loss on $\mathcal{D}_c \cup \mathcal{D}_p$ compared to any classifier $\theta \in \Theta$, the best lower bound is the supremum over all models in the model space $\Theta$. $\square$

**Corollary 4.2.1.** *If we further assume bi-directional closeness in the loss-based distance, we can also derive the lower bound on number of poisoning points needed to induce models that are $\epsilon$-close to the target model. More precisely, if $\theta_1$ being $\epsilon$-close to $\theta_2$ implies that $\theta_2$ is also $k \cdot \epsilon$ close to $\theta_1$, then we have,*

$$\sup_{\theta} z'(\theta) = \frac{L(\theta_p; \mathcal{D}_c) - L(\theta; \mathcal{D}_c) - NC_R \cdot R^* - Nk\epsilon}{\sup_{x,y} \left(l(\theta; x, y) - l(\theta_p; x, y)\right) + C_R \cdot R^* + k\epsilon}.$$

*where $R^*$ is an upper bound on the nonnegative regularizer $R(\theta)$.*

---

[2]In practice, the data space $\mathcal{X}$ is a closed convex set and hence, we can find $(x^*, y^*)$ using convex optimization. In other words, as we saw in experiments, calculating the lower bound is possible in practical scenarios.

*Proof of Corollary 4.2.1.* The lower bound for all $\epsilon$-close models to the target classifier is given exactly as follows:

$$\inf_{\|\theta' - \theta_p\|_{\mathcal{D}_{l,\mathcal{X},\mathcal{Y}}} \leq \epsilon} \sup_{\theta} \left( z(\theta, \theta') = \frac{L(\theta'; \mathcal{D}_c) - L(\theta; \mathcal{D}_c) + NC_R(R(\theta') - R(\theta))}{\sup_{x,y} \left( l(\theta; x, y) - l(\theta'; x, y) \right) + C_R(R(\theta) - R(\theta'))} \right),$$

where $\inf_{\|\theta' - \theta_p\|_{\mathcal{D}_{l,\mathcal{X},\mathcal{Y}}} \leq \epsilon}$ denotes $\theta'$ is $\epsilon$-close to $\theta_p$ in the loss-based distance. However, the formulation above is a min-max optimization problem and hard to analytically compute the lower bound (by plugging the lower bound formula into Algorithm 1. Therefore, we need to make several relaxations such that the lower bound is computable. For any model $\theta'$ that is $\epsilon$-close to $\theta_p$, based on the bi-directional assumption, then $\theta_p$ is $k\epsilon$-close to $\theta'$. Therefore we have,

$$L(\theta'; \mathcal{D}_c) - L(\theta; \mathcal{D}_c) = L(\theta'; \mathcal{D}_c) - L(\theta_p; \mathcal{D}_c) + L(\theta_p; \mathcal{D}_c) - L(\theta; \mathcal{D}_c) \geq -Nk\epsilon + L(\theta_p; \mathcal{D}_c) - L(\theta; \mathcal{D}_c)$$

and

$$\sup_{x,y} \left( l(\theta; x, y) - l(\theta', x, y) \right) = \sup_{x,y} \left( l(\theta; x, y) - l(\theta_p, x, y) \right) + \sup_{x,y} \left( l(\theta_p, x, y) - l(\theta'; x, y) \right)$$

$$\leq \sup_{x,y} \left( l(\theta; x, y) - l(\theta_p, x, y) + k\epsilon \right)$$

and the inequalities are all based on the definition of $\theta_p$ being $k\epsilon$-close to $\theta'$.

Plugging the above inequalities into the formula of $\sup_{\theta, \theta'}$ for model $\theta'$, and with the assumption that $0 \leq R(\theta) \leq R^*, \forall \theta \in \Theta$, we immediately have

$$\sup_{\theta} z(\theta, \theta') \geq \sup_{\theta} \frac{L(\theta_p; \mathcal{D}_c) - L(\theta; \mathcal{D}_c) - Nk\epsilon + NC_R(R(\theta') - R(\theta))}{\sup_{x,y} \left( l(\theta; x, y) - l(\theta_p; x, y) \right) - k\epsilon + C_R(R(\theta) - R(\theta'))}$$

$$\geq \sup_{\theta} \left( \frac{L(\theta_p; \mathcal{D}_c) - L(\theta; \mathcal{D}_c) - Nk\epsilon - NC_R \cdot R^*}{\sup_{x,y} \left( l(\theta; x, y) - l(\theta_p; x, y) \right) - k\epsilon + C_R \cdot R^*} = z'(\theta) \right).$$

Since the inequality holds for any $\theta'$, we have

$$\inf_{\|\theta' - \theta_p\|_{\mathcal{D}_{l,\mathcal{X},\mathcal{Y}}} \leq \epsilon} \sup_{\theta} z(\theta, \theta') \geq \sup_{\theta} z'(\theta)$$

and hence $z'(\theta)$ is a valid lower bound.

$\square$

**Remark 2** (Improving Results in Corollary 4.2.1). *Assuming $0 \leq R(\theta) \leq R^*$ is not a strong assumption and actually can be satisfied by many common convex models. For example, for SVM model with $\ell_2$-regularizer (in fact, applies to any regularizer $R(\theta)$ with $R(\mathbf{0}) = 0$), we have $R(\theta) \leq \frac{1}{C_R}$ and hence $R^* \leq \frac{1}{C_R}$. Moreover, we can further tighten the lower bound by better bounding the term $R(\theta') - R(\theta)$. Specifically, $R(\theta') - R(\theta) = R(\theta') - R(\theta_p) + R(\theta_p) - R(\theta)$ and we only need to have a tighter upper and lower bounds on $R(\theta') - R(\theta_p)$ utilizing some special properties of the loss functions. For the constant $k$ in the bi-directional closeness, we can also compute its value for some specific loss functions. For example, for Hinge loss, we can compute the value based on Corollary B.2.1 in Appendix B.*

# B    RELATING CLOSENESS OF LOSS-BASED DISTANCE TO CLOSENESS OF PARAMETERS

In theorem below, we show how one can relate the notion of $\epsilon$-closeness in Definition 1 in the main paper to closeness of parameters in the specific setting of hinge loss. We use this just as an example to show that our notion of $\epsilon$-closeness can be tightly related to the closeness of the models.

**Theorem B.1.** *Consider the hinge loss function $l(\theta; x, y) = \max(1 - y \cdot \langle x, \theta \rangle, 0)$ for $\theta \in \mathbb{R}^d$ and $x \in \mathbb{R}^d$ and $y \in \{-1, +1\}$. For $\theta, \theta' \in \mathbb{R}^d$ such that $\|\theta\|_1 \leq r$ and $\|\theta'\|_1 \leq r$, if $\theta$ is $\epsilon$-close to $\theta'$ in the loss-based distance, then, $\|\theta - \theta'\|_1 \leq r \cdot \epsilon$.*

**Remark 3.** *In Theorem B.1 above with $\ell_2$-regularizer, an upper bound on the $\ell_1$-norm of $\theta$ and $\theta'$ is $\sqrt{d/C_R}$. however, the models that we care about in practice usually have smaller norms.*

Remark 3 can be obtained by plugging $\mathbf{0} \in \mathbb{R}^d$ and compare the resulting (regularized) optimization loss to the model $\theta^*$ that minimizes the model loss.

*Proof of Theorem B.1.* We construct a point $x^*$ as follows:

$$x_i^* = \begin{cases} -\frac{1}{r}, & \text{if } \theta_i > \theta_i', i \in [d] \\ +\frac{1}{r} & \text{if } \theta_i \leq \theta_i', i \in [d] \end{cases}$$

Then we have

$$\langle \theta - \theta', x^* \rangle = \frac{1}{r} \cdot \|\theta - \theta'\|_1 \tag{3}$$

Since $\|\theta\|_1 \leq r$ we have

$$\langle x^*, \theta \rangle \geq -1 \tag{4}$$

and similarly since $\|\theta'\|_1 \leq r$ we have

$$\langle x^*, \theta' \rangle \geq -1. \tag{5}$$

Therefore by Inequalities equation 4 and equation 5 we have

$$l(\theta; x^*, -1) - l(\theta'; x^*, -1) = \max(1 + \langle x^*, \theta \rangle, 0) - \max(1 + \langle x^*, \theta' \rangle, 0) = \langle \theta - \theta', x^* \rangle$$

which by equation 3 implies

$$l(\theta; x^*, -1) - l(\theta'; x^*, -1) = \frac{1}{r} \cdot \|\theta - \theta'\|_1. \tag{6}$$

Now since we know that, $\forall x \in \mathbb{R}^d$, the loss difference between $\theta$ and $\theta'$ is bounded by $\epsilon$, the bound should also hold for the point $(x^*, -1)$, meaning that

$$\frac{1}{r} \cdot \|\theta - \theta'\|_1 \leq \epsilon.$$

which completes the proof. $\qquad\square$

**Theorem B.2.** *Consider the hinge loss function $l(\theta; x, y) = \max(1 - y \cdot \langle x, \theta \rangle, 0)$ for $\theta \in \mathbb{R}^d$ and $x \in \mathbb{R}^d$ and $y \in \{-1, +1\}$. For $\mathcal{X} = \{x \in \mathbb{R}^d \colon \|x\|_1 \leq q\}$ and $\mathcal{Y} = \{-1, +1\}$, For any two models $\theta, \theta'$ if $\|\theta - \theta'\|_1 \leq \epsilon$, then $\theta$ is $q \cdot \epsilon$-close to $\theta'$ in the loss-based distance. Namely,*

$$D_{\ell, \mathcal{X}, \mathcal{Y}}(\theta, \theta') \leq q \cdot \epsilon.$$

*Proof.* For any given $\theta$ and $\theta'$, by triangle inequality for maximum, we have

$$l(\theta; x, y) - l(\theta', x, y) = \max(1 - y \cdot \langle x, \theta \rangle, 0) - \max(1 - y \cdot \langle x, \theta' \rangle, 0) \leq \max(0, \langle yx, \theta' - \theta \rangle).$$

Therefore, we have

$$\max_{(x,y) \in \mathcal{X} \times \mathcal{Y}} l(\theta; x, y) - l(\theta'; x, y) \leq \max_{(x,y) \in \mathcal{X} \times \mathcal{Y}} \max(0, \langle yx, \theta' - \theta \rangle).$$

Our goal is then to obtain an upper bound of $O(\epsilon)$ for $\max_{(x,y) \in \mathcal{X} \times \mathcal{Y}} \langle yx, \theta' - \theta \rangle$ when $\|\theta - \theta'\|_1 \leq \epsilon$. To maximize $\langle yx, \theta' - \theta \rangle$ by choosing $x$ and $y$, we only need to ensure that $\text{sign } yx_i = \text{sign } \theta_i, i \in [d]$. Therefore, based on the assumption that $\frac{1}{q}\|x\| \leq 1$ (i.e., $\frac{1}{q}|x_i| \leq 1, i \in [d]$) we have

$$\max_{(x,y) \in \mathcal{X} \times \mathcal{Y}} \frac{1}{q} \langle yx, \theta' - \theta \rangle = \sum_{i=1}^d \frac{1}{q}|x|_i|\theta_i - \theta_i'| \leq \sum_{i=1}^d |\theta_i - \theta_i'| = \|\theta - \theta'\|_1 \leq \epsilon,$$

which concludes the proof. $\qquad\square$

**Corollary B.2.1.** *For Hinge loss, with Theorem B.1 and Theorem B.2, if $\theta$ is $\epsilon$-close to $\theta'$, then $\theta'$ is $r \cdot q \cdot \epsilon$-close to $\theta$.*

## C  Instantiating Theorem 4.1 for the case of SVM

Here we show how to instantiate Theorem 4.1 for SVM with exact constants instead of the asymptotic notations. We need to calculate the constant $\gamma$ to get the exact constant. Imagine the feature domain is $\mathbb{R}^d$. Now we calculate the constant $C$ as follows. Let $i_\theta^* = \arg\min_{i\in[d]} |\theta[i]/\theta_p[i]|$ and $\alpha_\theta = |\theta[i_\theta^*]/\theta_p[i_\theta^*]|$. Let $x_\theta^* \in \mathbb{R}^d$ be a point where is equal to 0 everywhere and is equal to $1/\theta_p[i_\theta^*]$ on the $i^*$ coordinate. We have,

$$l(\theta, x_\theta^*, +1) - l(\theta_p, x_\theta^*, +1) = l(\theta, x_\theta^*, +1) \geq (1 - \alpha_\theta). \tag{7}$$

Now we can calculate $C$ as follows

$$C = \left( \inf_{\substack{\theta \in F \\ \text{s.t. } R(\theta_p) > R(\theta)}} \sup_{x,y} (l(\theta; x, y) - l(\theta_p; x, y))/(R(\theta_p) - R(\theta)) \right)$$

$$\geq \left( \inf_{\substack{\theta \in F \\ \text{s.t. } R(\theta_p) > R(\theta)}} (l(\theta, x_\theta^*, +1) - l(\theta_p, x_\theta^*, +1))/(R(\theta_p) - R(\theta)) \right)$$

$$\text{(By Inequality 7)} \quad \geq \inf_{\substack{\theta \in F \\ \text{s.t. } R(\theta_p) > R(\theta)}} \sup_{x,y} \frac{1 - \alpha_\theta}{R(\theta_p) - R(\theta)}$$

$$\text{(By definition of } \alpha_\theta) \quad \geq \inf_{\substack{\theta \in F \\ \text{s.t. } R(\theta_p) > R(\theta)}} \frac{1 - \alpha_\theta}{R(\theta_p)(1 - \alpha_\theta^2)}$$

$$\geq \inf_{\substack{\theta \in F \\ \text{s.t. } R(\theta_p) > R(\theta)}} \frac{1 - \alpha_\theta}{R(\theta_p)(1 - \alpha_\theta^2)}$$

$$\geq \frac{1}{2R(\theta_p)}$$

Therefore $\gamma \geq 1 - 2 \cdot C_R \cdot R(\theta_p)$. On the other hand, we can also calculate $\alpha(T)$ based on the exact form given in the proof of Theorem 4.1.

## D  Indiscriminate Setting Experiments

In this section, we evaluate the attacks in the conventional indiscriminate attack setting, where the attacker's goal is just to reduce the overall accuracy of the model.

**Datasets and Models.** For the indiscriminate attack, we use the MNIST 1–7 dataset, which consists of the digits 1 and 7 and is commonly used for evaluating indiscriminate poisoning attacks against binary classification (Steinhardt et al., 2017; Biggio et al., 2012; Xiao et al., 2012). MNIST 1–7 contains 13,007 training and 2,163 test samples. The dataset contains 784 features and all the features are normalized into range $[0, 1]$. For completeness, the Adult dataset used for subpopulation attack is downsampled to form a class-balanced dataset and contains 15682 training data and 7692 test data. The dataset contains 57 features and the features are also normalized into range $[0, 1]$ (except for the binary features). All of the processed datasets are included in the supplementary material. We still adopt linear SVM model in the indiscriminate attack scenario. All of the models for both datasets set the regularization parameter $C_R = 0.09$. The clean accuracy of SVM model on MNIST 1–7 is 98.9% and the accuracy on Adult dataset is 78.5%.

**Target Classifiers.** Accuracy of the clean MNIST 1–7 model has around 1% error rate on the test set. For our experiment, we aim to generate three target classifiers with overall test errors around 5%, 10% and 15%. To generate target classifiers with desired error rates, we follow the heuristic strategy proposed by Koh et al. (2018) to generate multiple candidate target classifiers, and then among all the valid candidate models that satisfy the error rate requirement we choose the one with lowest loss on the clean training set. Using this approach, the final target classifiers induced have overall test accuracy of 94.0%, 88.8% and 83.3% respectively. (We describe a better way of finding the target classifiers in Appendix D.1, but for comparison purposes do not use those in the results here.)

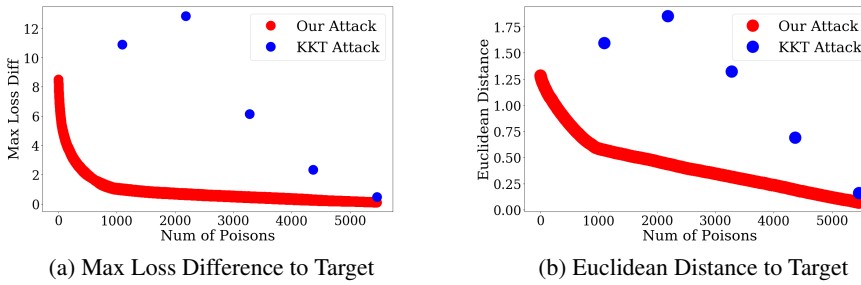

(a) Max Loss Difference to Target

(b) Euclidean Distance to Target

Figure 3: Attack convergence (results shown are for the target classifier of error rate 10%). The maximum number of poisons is set using the 0.1-close threshold to target classifier

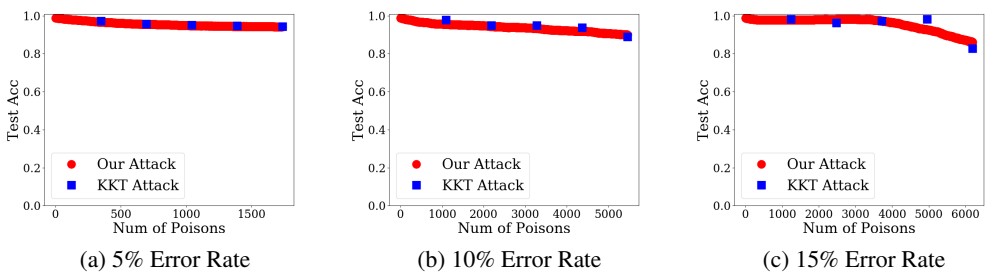

(a) 5% Error Rate

(b) 10% Error Rate

(c) 15% Error Rate

Figure 4: Test accuracy for each target model of given error rate with classifiers induced by poisoning points obtained from our attack and the KKT attack.

**Convergence.** We show the convergence of Algorithm 1 by reporting the maximum loss difference and Euclidean distance between the classifier induced by the attack and the target classifier. Figures 3a and 3b summarize the results for the target classifier with a 10% error rate. The maximum number of poisoning points in the figure is obtained when the classifier from Algorithm 1 is 0.1-close to the target classifier in the loss-based distance. In Figure 3, the classifier induced by our algorithm steadily converges to the target classifier both in the maximum loss difference and Euclidean distance, while the classifier induced by the KKT attack diverges initially and then starts to converge to the target model. At the maximum number of points, the maximum loss difference of KKT-induced classifier to the target is 0.46, compared to 0.1 for the classifier induced by our attack. For the Euclidean distance, the KKT-induced classifier is 0.16 away, compared to 0.07 for the classifier induced by our attack.

**Attack Success.** We next compare the classifier induced from our attack to the classifier induced by the KKT attack in terms of their overall test accuracy. Similarly, the maximum number of poisoning points in Figure 4 is obtained by running our attack with 0.1-closeness (in loss-based distance) to the target as the input. In terms of the test accuracy, our attack has a comparable attack success rate compared to the KKT attack. Specifically, for target models of 5% and 10% error rates, both methods have almost identical performance, as shown in Figures 4a and 4b. For the target model of 15% error rate (test accuracy is 83.3%), the KKT attack is more successful than our attack, inducing models with 82.7% accuracy (17.3% attack success rate) compared to 85.9% accuracy (14.1% attack success rate) for our attack. Interestingly, in this case, the performance of models induced by the two attacks with fewer poisoning points our attack results in models with lower test accuracy (higher attacker success) than the KKT attack. To summarize, for the indiscriminate scenario, our attack produces classifiers that have much closer distance to the target models than the KKT attack, and has comparable attack success rates with the KKT attack.

**Lower Bound on Number of Poisons.** We next check the optimality of our attack in the indiscriminate attack scenario. Similar to the subpopulation attack setting, we still use Theorem 4.2 to compute the lower bound of the induced classifier from our attack by using it as the input to Algorithm 1, and terminating when the induced classifier is 0.1-close to the given target model. In Table 2, our

|  | 5% Error | 10% Error | 15% Error |
|---|---|---|---|
| # of Poisons | 1737 | 5458 | 6192 |
| Lower Bound | 874 | $3850.4 \pm 0.8$ | 4904 |

Table 2: Poisoning points needed to achieve target classifiers induced from our attack. Top row means number of poisoning points used by our attack. Bottom row means the lower bound computed from Theorem 4.2 for the target classifiers. All results are averaged over four runs. An integral value in a cell means we get exactly that same value for all four runs; for the one cell where we observe variation, we report the average and standard error.

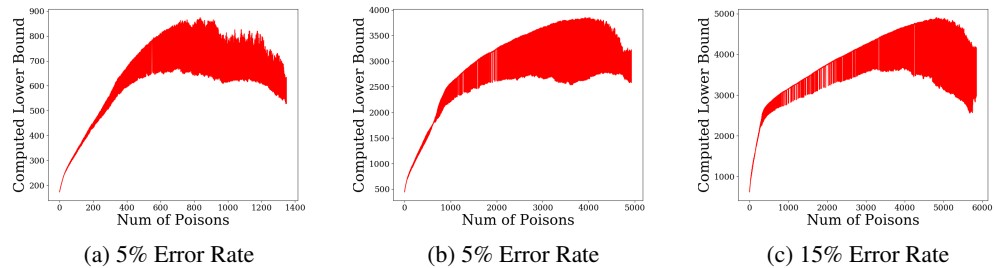

(a) 5% Error Rate    (b) 5% Error Rate    (c) 15% Error Rate

Figure 5: Lower bound computed in each iteration of running algorithm 1 when the target classifier of the algorithm is the classifier induced from our Attack (classifier in Table 2). The maximum number of poisons is set using the $0.1$-close threshold to classifier induced from our attack.

|  | 5% Error | 10% Error | 15% Error |
|---|---|---|---|
| # of Poisons | 1737 | 5458 | 6192 |
| Lower Bound | 856 | 4058.4+1.4 | 5031.4+4.8 |

Table 3: Poisoning points needed to achieve target classifiers induced from the KKT attack. Top row means number of poisoning points used by the KKT attack. Bottom row means the lower bound computed from Theorem 4.2 for the target classifiers. All results are averaged over 4 runs, integer value in the cell means we get exactly same value for 4 runs and others are shown with the average and standard error.

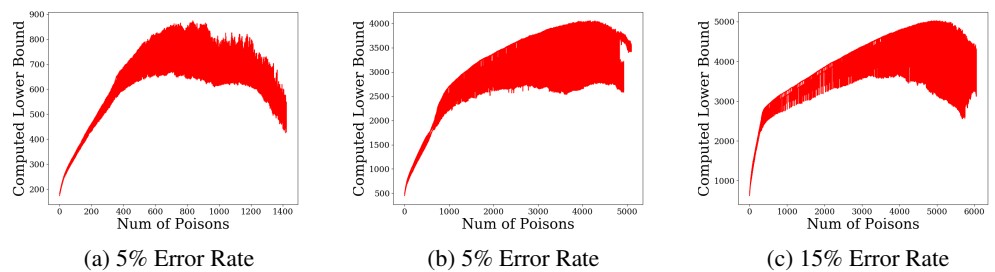

(a) 5% Error Rate    (b) 5% Error Rate    (c) 15% Error Rate

Figure 6: Lower bound computed in each iteration of running algorithm 1 when the target classifier of the algorithm is the classifier induced from the KKT Attack (classifier in Table 3). The maximum number of poisons is set using the $0.1$-close threshold to KKT induced classifier.

calculated lower bound shows that there exists a relatively large gap between the number of poisoning points, especially for the induced classifier from our attack for the target model of 5% error rate, where the lower bound is only 50% of the actual number of poisoning points used. For the induced classifier for the target model of 15% error rate, the gap between the number of poisoning points and the lower bound is smallest, with the lower bound taking 79% of the number of poisoning points.

| Target Models | Test Acc (%) | | Loss on Clean Set | | # of Poisons | |
|---|---|---|---|---|---|---|
| | Original | Improved | Original | Improved | Original | Improved |
| 5% Error | 94.0 | 94.9 | 2254.6 | 1767.1 | 2170 | 1340 |
| 10% Error | 88.8 | 88.9 | 4941.0 | 3233.1 | 5810 | 2432 |
| 15% Error | 83.3 | 84.5 | 5428.4 | 4641.6 | 6762 | 3206 |

Table 4: Comparison of two target generation methods on number of poisoning points used to reach 0.1-closeness to the target. *Original* indicates the original target generation process from Koh et al. (2018). *Improved* denotes our improved target generation process with adaptive model updating.

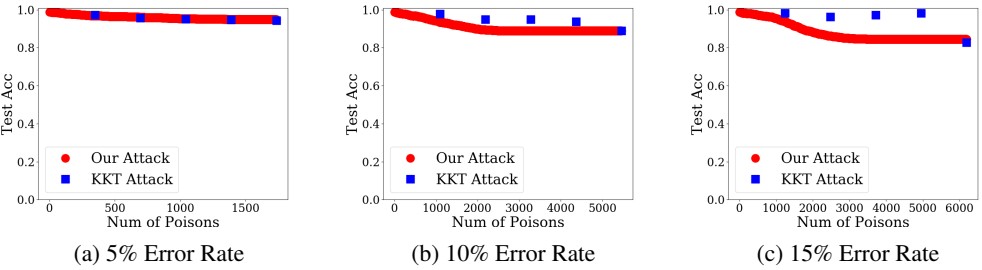

(a) 5% Error Rate          (b) 10% Error Rate          (c) 15% Error Rate

Figure 7: Test accuracy with classifiers obtained from our attack and KKT attack. Target model for KKT attack is generated from the original generation process and target model for our attack is generated from the improved generation process. Maximum number of poisoning points is obtained by running our attack with target model generated from the original process and resultant classifier is 0.1-close to the target.

The relatively large gap indicates that either the estimated lower bound is not tight or the attack itself is not close to optimal. To gain more insights into this problem, we further show the computed lower bound at each iteration when running Algorithm 1 and Figure 5 summarizes the results. From the Figure, we see that, the peak value of the curve (i.e., highest lower bound) always appears before the termination of the algorithm, indicating that the computed lower bound is likely to be tight and we may need to further improve the attack algorithm.

For completeness, we also repeat the lower bound computation process for classifiers induced from the KKT attack. The KKT induced classifiers are obtained by running the KKT attack with target classifiers of different error rates as target input. The target number of poisoning points of KKT attack is given by the size of poisoning set returned from our attack when our algorithm terminates when the induced classifier is 0.1-close to the target model of different error rates. Then the lower bound computation process is identical to the above – we simply send the KKT induced classifier as target input to Algorithm 1 and terminate it when the induced classifier from our algorithm is 0.1-close to the given target model (i.e., KKT induced classifier). The results are summarized in Table 3 and we observe that there also exists a relatively large gap between the lower bound and the number of poisoning points used by the attack. Similarly, we also plot the lower bound computed at different iterations of Algorithm 1 in Figure 6, and find that the peak value also appears before the termination of the algorithm, indicating that the lower bound might be tight and we need a stronger attack strategy to close the gap.

## D.1 IMPROVED TARGET GENERATION PROCESS

The original heuristic approach works by finding different quantiles of training points that have higher loss on the clean model, flipping their labels, repeating those points for multiple copies, and adding them to the clean training set. We find that, in the process of trying different quantiles and copies of high loss points, if we also adaptively update the model where the high loss points are found (instead of just always fixing it to be the clean model), we can generate a valid target classifier with even lower loss on the clean training. Such an improved generation process can significantly reduce the number of poisoning points needed to reach the same $\epsilon$-closeness (with respect to the loss-based distance)

to the target classifier, consistent with the claims in Theorem 4.1 in the main paper. In addition, we find that, if we compare our attack with improved generation process to the KKT attack with the original generation process (Koh et al., 2018), we can also reach the desired target error rate much faster using our attack.

**Implication of Theorem 4.1.** We first empirically validate the implication of Theorem 4.1 in the main paper: to obtain the same $\epsilon$-closeness in loss-based distance, a target classifier with lower loss on the clean training set $\mathcal{D}_c$ requires fewer poisoning points. Therefore, when adversaries have multiple target classifiers that satisfy the attack goal, the one with lower loss on clean training set is preferred.

For both the original and improved target generation methods, we generate three target classifiers with error rates of 5%, 10% and 15%. The original target classifier generation method returns classifiers with test accuracy of 94.0%, 88.8% and 82.3% respectively (also used in the previous experiments of indiscriminate attack). The improved target generation process returns target classifiers with approximately the same test accuracy (94.9%, 88.9% and 84.5%). However, for classifiers returned from the two generation methods of same error rate, the improved generation method produces classifiers with significantly lower loss compared to the original generation approach.

Table 4 compares the two target generation approaches by showing the number of poisoning points needed to get $0.1$-close to the corresponding target model of same error rate. For example, for target models of 15% error rate, the model from the original approach has a total clean loss of 5428.4 while our improved method reduces it to 4641.6. With the reduced clean loss, getting $0.1$-close to the target model generated from our improved process only requires 3206 poisoning points, while reaching the same distance from the target model produced by the original method would require 6762 poisoning points, a more than 50% reduction.

**End-to-End Comparison.** Figure 7 compares the two attacks in an end-to-end manner in terms of their test accuracy. With the improved target generation process, our attack can achieve the desired error rate much faster than the KKT attack with the original process. For the KKT attack with target model generated from the original process, we determine the target number of poisoning points using the size of poisoning set returned from running Algorithm 1 with $0.1$-closeness and target model from original process as inputs. To run our attack with improved generation process, we terminate the algorithm when the size of the poisoning points is same as the number of poisoning points used by the KKT attack with original process. Such a termination criteria helps us to ensure that both attacks use same number of poisoning points and can be compared easily. We also evaluate the KKT attack on fractions of the maximum target number of poisoning points (0.2, 0.4, 0.6, and 0.8), as in the previous experiments. The accuracy plot shows that our attack (with improved target model) can achieve the desired error rate (e.g., 10%) much faster than the KKT attack (with original target model), especially for the target classifiers of error rates of 10% and 15%.

## E    COMPARISON OF MODEL-TARGETED AND OBJECTIVE-DRIVEN ATTACKS

Although model-targeted attacks work to induce the given target classifiers by generating poisoning points, the end goal is still to achieve the attacker objectives encoded in the target models. In terms of the comparison to the objective-driven attacks, we first demonstrate that, objective-driven attacks can be used to generate a target model, which can then be used as the target for a model-targeted attack, resulting in an attack that achieves the desired attacker objective with fewer poisoning points. Then, we show that in order to have competitive performance against state-of-the-art objective-driven attacks (e.g., the min-max attack (Steinhardt et al., 2017)), the target classifiers should be generated carefully, such that the attacker objectives of the target classifiers can be achieved efficiently with model-targeted attacks using fewer poisoning points. Although the investigation of a systematic approach to generate such "desired" classifiers is out of the scope of this paper, in the indiscriminate setting, we have some empirical evidence. Specifically, we find that target classifiers with lower loss on clean training set and higher error rates (higher than what are desired in the attacker objectives) often require fewer poisoning points to achieve the attacker objectives. The following experiments are conducted on the MNIST 1–7 dataset.

**Target Models Generated from Objective-driven Label-Flipping Attacks.** In our experiments, the target classifiers are generated from the label flipping based objective driven attacks that are

effective but need too many poisoning points to achieve their objective. Then, our attacks are deployed to achieve the same objective with fewer poisoning points. Table 5 shows the number of poisoning points used by the label-flipping attack and our model-targeted attack, to achieve desired attack objectives of increasing the test error to certain amount. We can see that, using our attack, the number of poisoning points used by label-flipping attacks can be saved up to 73%.

**Comparison to the Min-Max Attack.** Still using target classifiers generated from label-flipping attacks, we show that our attack can outperform the state-of-the-art min-max attack (Steinhardt et al., 2017) at reducing the overall test accuracy, under same amount of poisoning points. Since we aim to produce target classifiers with lower loss on clean training set and higher error rates, we adopt the improved target model generation process described in Section D.1 (helps to reduce the loss on clean training set) and generate a classifier of 15% error rate. With the target model, we terminate our attack when the induced model is 0.1-close to the target model in terms of the loss-based distance. However, to we compare our attack to the min-max attack conveniently, we compare their accuracy reduction at different poisoning ratios (i.e., 5%, 15% and 30%), which is the common evaluation strategy of objective-driven attacks in the indiscriminate setting (Biggio et al., 2011; Steinhardt et al., 2017; Koh et al., 2018). Table 6 summarizes the results. From the table, we observe that, compared to the min-max attack, our attack reduces more on the test accuracy under the same poisoning budget and the gap becomes larger when the poisoning ratio increases.

**Comparison to the Label-Flipping Subpopulation Attack.** We also compare our attack to the label-flipping subpopulation attack from Jagielski et al. (2019). This attack works by randomly sampling fixed number (constrained by the poisoning budget) of instances from the training data of the subpopulation, flipping their labels and then injecting them into to the original training set. Although this attack is very simple, it shows relatively high attack success when the goal is to cause misclassification on the selected subpopulation (Jagielski et al., 2019).

To be consistent with our experiments in Section 5, we assume the attacker objectives are still to induce a model that has 0% accuracy on a selected subpopulation. For each of the SVM and logistic regression models, we selected the three subpopulations with highest test accuracy (all end up having have 100% accuracy). In indiscriminate setting, we already observed that models with lower loss on clean training set and larger overall error rates can achieve attacker objectives of smaller error rates faster. However, to leverage this observation into our subpopulation experiments, one challenge is the attacker objective is to have 100% test error on the subpopulation, but no classifiers can have test errors larger than 100%. To tackle this, we select models with larger loss on training samples from the subpopulation, with a hope that this process is "equivalent" to selecting target models with larger error rates (on subpopulation) than 100%. To this end, we heuristically select targeted models that satisfy the attacker objective, have larger loss on the training data from the subpopulation, and have relatively low loss on the entire clean training set. Empirically, this selection strategy works better than the original target generation process (as done in Section 5) in achieving the attacker objectives. A more detailed and systematic investigation of the target model search process is left as the future work.

To check the effectiveness of achieving the attacker objectives, we first run our attack and terminate when our attack achieves the attacker objective to have 0% accuracy on the selected subpopulation, and record the number of poisoning points used. Then, we run the random label-flipping attack with the same number of poisoning points. For both attacks, we report the final test accuracies of the resulting models on the subpopulations.

| Attacker Objectives | 5% Error | 10% Error | 15% Error |
|---|---|---|---|
| Label-flipping Attack | 6,510 | 8,648 | 10,825 |
| Our Attack | 1,737 | 5,458 | 6,192 |

Table 5: Generate target classifiers using objective-driven label-flipping attacks and achieve similar attacker objectives using our attack with fewer poisoning points. The attacker objectives are to increase the test error to certain amounts (i.e., 5%, 10% and 15%) and the target classifiers to our attack are generated by running the label-flipping attacks with given attacker objectives.

| Poisoning Ratio | 5% | 15% | 30% |
|---|---|---|---|
| Min-max Attack | 97.0% | 93.9% | 92.9% |
| Our Attack | 96.2% | 88.6% | 85.0%* |

Table 6: Comparison of our attack to the min-max attack with different poisoning ratios. The target model of our attack is of 15% error rate. The poisoning ratio is with respect to the full training set size of 13,007. Each cell in the table denotes the test accuracy of the classifier after poisoning. The clean test accuracy is 98.9%. Our attack at 30% poisoning ratio is marked with "*" because the attack terminates when the induced model is 0.1-close to the target model, which only uses 2,894 poisoning points and is less than the 30% ratio.

| | SVM | | | Logistic Regression | | |
|---|---|---|---|---|---|---|
| | Cluster 0 | Cluster 1 | Cluster 2 | Cluster 0 | Cluster 1 | Cluster 2 |
| Our Attack | 0% | 0% | 0% | 0% | 0% | 0% |
| Label-Fipping | 31.4% | 2.8% | 15.5% | 15.9% | 14.0% | 19.1% |

Table 7: Comparison of our attack to the label-flipping based subpopulation attack. The table compares the test accuracy on subpoplation of Adult dataset under same number of poisning points. The number of poisons are determined when our attack achieves 0% test accuracy on the subpopulation. Cluster 0-3 in the logistic regression and SVM models denote different clusters. For logistic regression, number of poisoning points for Cluster 0-3 are 1,575, 1,336 and 1,649 respectively. For SVM, number of poisoning points for Cluster 0-3 are 1,252, 1,268 and 1,179 respectively.

The attack comparisons on different subpopulation clusters and models are given in Table 7. Results in the table compare our attack and the label-flipping attack over the three distinct subpopulation clusters for the SVM and logistic regression models. Across all settings, our attack is considerably more successful. The number of poisoning points needed to reach the 0% accuracy goal is small compared to the entire training set size (e.g., the maximum poisoning ratio is only 10.5%). The gap between our attack and the label-flipping attack is fairly small. For example, for Cluster 1 in the SVM experiment, the label-flipping attack is also quite successful and reduces the test accuracy to 2.8% (our attack achieves 0% accuracy). We believe the success of label-flipping attack is due to the following two reasons. First, label-flipping in the subpopulation setting can be successful because smaller subpopulations show some degree of locality and hence, injecting points (from the subpopulation) with flipped labels can have a strong impact on the selected subpopulation. This is confirmed by empirical evidence that increasing the subpopulation size (i.e., reducing its locality) gradually reduces the label-flipping effectiveness and the attack becomes much less effective in the indiscriminate setting (i.e., subpopulation is the entire population). Second, the Adult dataset only contains 57 features, where 53 of them are binary features with additional constraints. Therefore, the benefit from optimizing the feature values is less significant as the optimization search space of our attack is fairly limited.

# F  ADDITIONAL EXPERIMENTS

In this section, we provide the results of evaluating our attack and the KKT attack on SVM models for an additional dataset (Section F.1) and on logistic regression models for three datasets (Section F.2). The results here are consistent with our findings in on the datasets and models in Section 5, but provide further evidence of the general effectiveness of our attack.

## F.1  ATTACKS ON SVM TRAINED ON DOGFISH

In this section, we introduce the results of SVM model evaluated on the Dogfish dataset.

**Dataset.** The Dogfish dataset contains *dog* and *fish* images of dimensions $298 \times 298 \times 3$. This dataset has been used as a binary classification task by by previous works in evaluating poisoning attacks in the indiscriminate setting (Koh & Liang, 2017; Steinhardt et al., 2017; Koh et al., 2018).

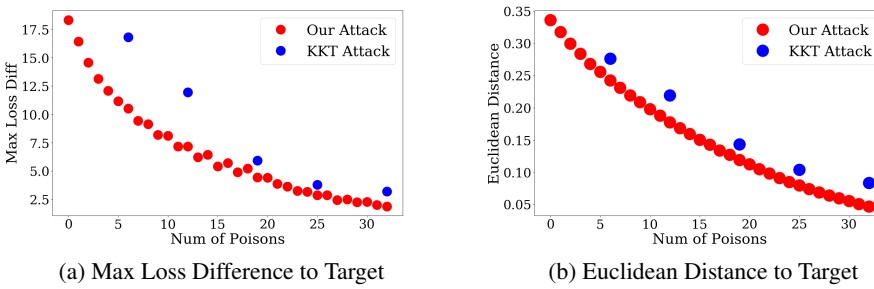

(a) Max Loss Difference to Target

(b) Euclidean Distance to Target

Figure 8: SVM on Dogfish dataset: attack convergence (results shown are for the target classifier of error rate 10%). The maximum number of poisons is set using the 2.0-close threshold to target classifier

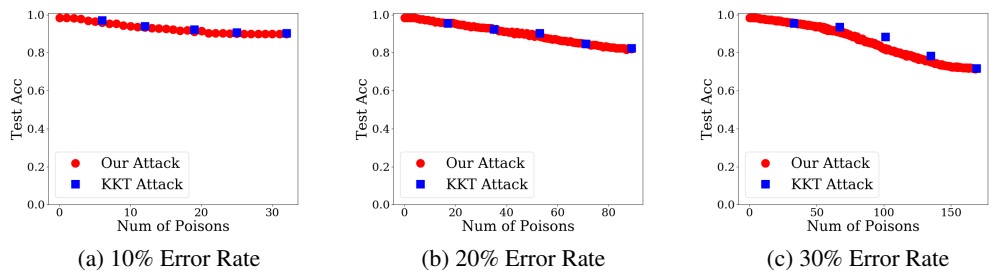

(a) 10% Error Rate

(b) 20% Error Rate

(c) 30% Error Rate

Figure 9: SVM on Dogfish dataset: test accuracy of each target model of given error rate with classifiers induced by poisoning points obtained from our attack and the KKT attack.

|  | 10% Error | 20% Error | 30% Error |
|---|---|---|---|
| Poisoning Points | 32 | 89 | 169 |
| Lower Bound | 13 | 36 | 67 |

Table 8: SVM on Dogfish dataset: poisoning points needed to achieve target classifiers induced from our attack. Top row means number of poisoning points used by our attack. Bottom row means the lower bound computed from Theorem 4.2 for the induced classifiers.

To make classification easy for linear models, both Steinhardt et al. (2017) and Koh et al. (2018) use the extracted features (of the original images) from the ImageNet Inception model (Szegedy et al., 2016) and then apply linear models to complete the classification task. We also adopt the extracted features for classification, so each instance has 2,048 features. The Dogfish dataset has 1,800 training samples and 600 test samples. As in the previous work, we conduct a conventional indiscriminate attack on it, where the adversary's goal is to reduce the overall test accuracy.

**Target Classifiers.** The clean accuracy of the SVM model on the Dogfish dataset is 98.5%. We generate target classifiers of overall test error rates around 10%, 20% and 30% using a similar (heuristic) target generation process (Koh et al., 2018) as in the case of MNIST 1–7 dataset. The final test accuracies of the obtained target classifiers are 89.3%, 78.3% and 67.2% respectively.

**Attack Results.** The convergence of our attack is demonstrated in Figure 8, both in the loss-based distance and the actual model distance in $\ell_2$-norm. The maximum number of poisons for the experiments is obtained when the classifier from Algorithm 1 is 2.0-close to the target classifier. Our attack steadily converges to the target model for both metrics, and has a faster convergence rate than the KKT attack. Similar observations are found in other attack settings, as summarized in Figure 9. Our attack is slightly more successful than the KKT attack in all three target classifiers we tested. For the models induced from our attack, in Table 8, we also show the gap between the number of

poisoning points of our attack and the theoretical lower bound. Although our attack can induce the target models using very few poisoning points (recall that the entire training set size is 1,800), there is still some gap to the theoretical lower bound. We also repeated the same process on the model induced from the KKT attack and again observe gaps between the number of poisoning points and the corresponding lower bound. These suggest that there is still potentially room for improvement in finding more efficient model-targeted poisoning attacks.

## F.2 ATTACKS ON LOGISTIC REGRESSION

In this section, we evaluate our attack on the logistic regression models for the Adult, MNIST 1–7 and Dogfish datasets.

The convergence guarantee in the paper also holds for logistic regression model (more generally, holds for any Lipschitz and convex function with strongly convex regularizer). However, for logistic regression, we may not be able to efficiently search for the globally optimal point with maximum loss difference (Line 4 in Algorithm 1) because the difference of two logistic losses is not concave. Therefore, we adopt gradient descent strategy, using the Adam optimizer (Kingma & Ba, 2014) to search for the point that (approximately) maximizes the loss difference. This is in contrast to the SVM model, where the difference of Hinge loss is piece-wise linear and we can deploy general (convex) solvers to search for the globally optimal point in each linear segment (Diamond & Boyd, 2016; Inc, 2020). However, as will be demonstrated next, poisoning points with approximate maximum loss difference can still be very effective. More formally, if the approximate maximum loss difference $\hat{l}$ found from local optimization techniques is within a constant factor from the globally optimal value $l^*$ (i.e., $\hat{l} \geq \alpha l^*, 0 < \alpha < 1$), then we still enjoy similar convergence guarantees. A similar issue of global optimality also applies to the KKT attack (Koh et al., 2018), where the attack objective function is no longer convex for logistic regression models, and therefore, we also utilize gradient based technique to (approximately) solve the optimization problem. Since the maximum loss difference found for logistic regression models may not be the globally optimal value, in these experiments we did not compute the lower bound (Theorem 4.2) on number of poisoning points to induce the poisoned model from our attack, which requires obtaining the actual maximum loss difference[3].

**Target Classifiers.** The clean accuracies of logistic regression models on the three datasets are 79.9% on Adult, 98.1% on MNIST 1–7 and 98.5% on Dogfish. Target classifiers for logistic regression models are generated similarly to their SVM counterpart on each dataset. For Adult dataset, the subpopulations are generated exactly the same as in the SVM case and form a total of 20 subpopulations, where instances in each subpopulation all belong to the "low-income" group. Among all the subpopulations, we select three with the highest test accuracy on the subpopulations and they all have 100% accuracy. The target classifiers are then generated to have 0% accuracy on the subpopulations using the target generation method described in Section 5. On the MNIST 1–7 dataset, target models are generated to have around 5%, 10% and 15% overall test errors and final test accuracies of the obtained target models are 94.7%, 89.0% and 84.5%. For Dogfish, target models are generated to have around 10%, 20% and 20% overall test errors and the final test accuracies of the resulting models are 89.0%, 79.5% and 67.3%.

**Results on Adult.** Figure 10 shows the effectiveness of our attack on logistic regression models trained on the Adult dataset, using the loss-based distance and the actual model distance in $\ell_2$-norm. The maximum number of poisons for the experiments is obtained when the classifier from Algorithm 1 is 0.05-close to the target classifier. Our attack steadily converges to the target model while the KKT attack fails to have a reliable convergence. Similar observations are also found in other attack settings, as shown in Figure 11. Our attack is much more successful than the KKT attack, especially for the attack on Cluster 1.

**Results on MNIST 1–7.** Figure 12 shows the convergance of our attack on LR models trained on the MNIST 1–7. The maximum number of poisons for the experiments is obtained when the classifier from Algorithm 1 is 0.1-close to the target classifier. Our attack converges to the target model while

---

[3]In the lower bound computation, when the loss difference is not concave, we can use a concave upper bound for it to obtain a valid lower bound and as long as the upper bound is relatively tight, the lower bound is still meaningful.

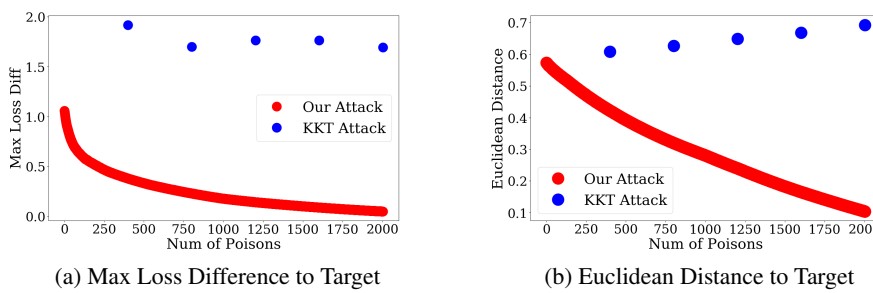

(a) Max Loss Difference to Target

(b) Euclidean Distance to Target

Figure 10: Logistic regression model on Adult dataset: attack convergence (results shown are for the first subpopulation, Cluster 0). The maximum number of poisons is set using the 0.05-close threshold to target classifier.

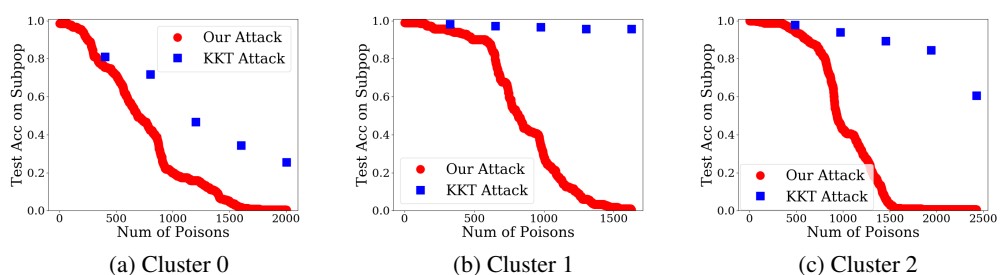

(a) Cluster 0

(b) Cluster 1

(c) Cluster 2

Figure 11: Logistic regression model on Adult dataset: test accuracy for each subpopulation with classifiers induced by poisoning points obtained from our attack and the KKT attack.

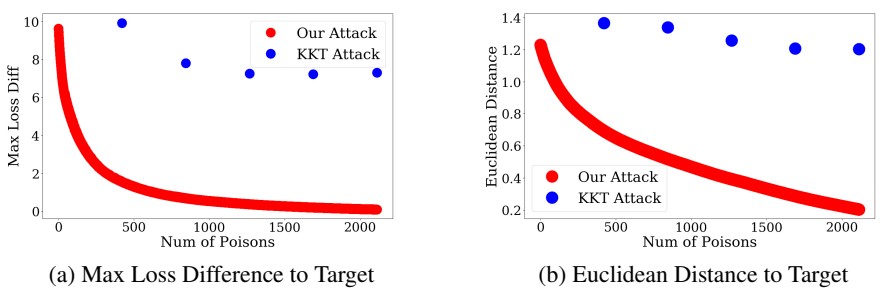

(a) Max Loss Difference to Target

(b) Euclidean Distance to Target

Figure 12: Logistic regression model on MNIST 1–7 dataset: attack convergence (results shown are for the target classifier of error rate 10%). The maximum number of poisons is set using the 0.1-close threshold to target classifier.

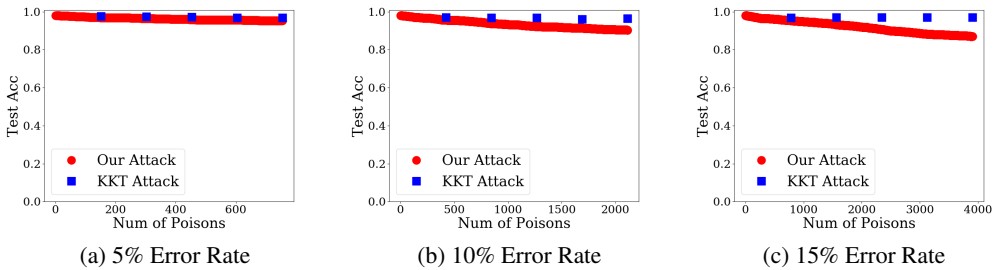

(a) 5% Error Rate

(b) 10% Error Rate

(c) 15% Error Rate

Figure 13: Logistic regression model on MNIST 1–7 dataset: test accuracy for each target model of given error rate with classifiers induced by poisoning points obtained from our attack and the KKT attack.

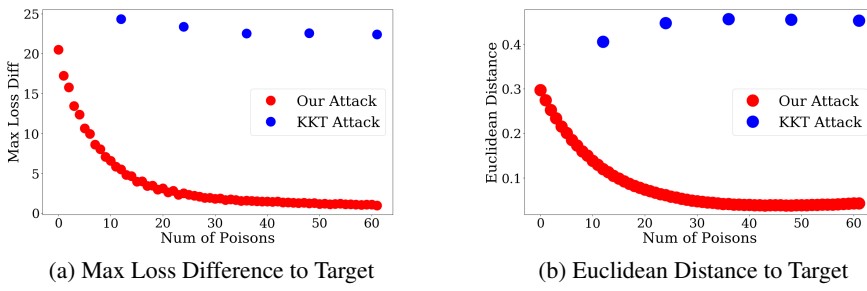

(a) Max Loss Difference to Target        (b) Euclidean Distance to Target

Figure 14: Logistic regression model on Dogfish dataset: attack convergence (results shown are for the target classifier of error rate 10%). The maximum number of poisons is set using the 1.0-close threshold to target classifier

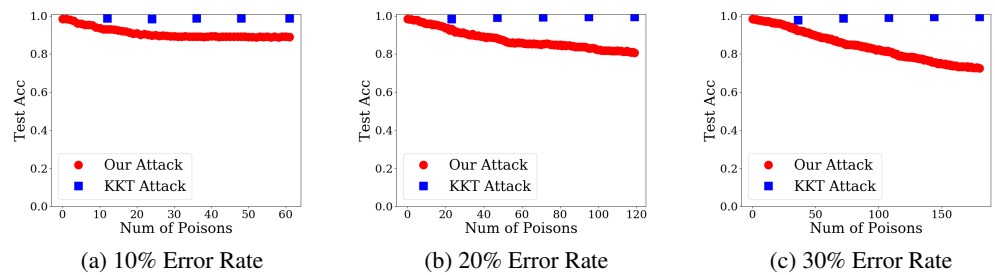

(a) 10% Error Rate     (b) 20% Error Rate     (c) 30% Error Rate

Figure 15: Logistic regression model on Dogfish dataset: test accuracy of each target model of given error rate with classifiers induced by poisoning points obtained from our attack and the KKT attack.

the KKT attack diverges. Similar observations are also found for other attack settings, as shown in Figure 13. In these settings, our attack is much more successful than the KKT attack. In fact, the KKT attack seems to not find a useful set of poisoning points as its induced models did not show a significant drop from the clean accuracy of 98.1%. We suspect this is due to the highly non-convex nature of the attacker objective[4] when attacking logistic regression models. In contrast, our attack only deals with maximizing the difference of two logistic losses, which is simpler than the KKT attacker objective, and results in a successful attack.

**Results on Dogfish.** Figure 14 shows attack results of logistic regression models on the Dogfish dataset. The maximum number of poisons for the experiments is obtained when the classifier from Algorithm 1 is 1.0-close to the target classifier. Our attack converges to the target model while the KKT attack fails to converge. Attack success comparisons are given in Figure 15. As with MNIST 1–7, our attack succeeds in settings where the KKT attack does not lead to significant accuracy drops from the clean accuracy of 98.5%. We believe this is also because of the highly non-convex nature of the KKT attack objective.

---

[4]The attacker objective is related to minimizing norm of the gradient, and becomes complicated for logistic regression models. Details of the attack formulation are in Koh et al. (2018).

