# OpenReview forum: "Model-Targeted Poisoning Attacks with Provable Convergence"
_ICLR.cc/2021/Conference — Reject_

### Official Review · AnonReviewer4 · 2020-10-27
**ill-defined problem, unjustified approach, and insufficient evaluation**

**Rating:** 3
**Confidence:** 4

**Review:**

Most existing data poisoning attacks are objective driven, e.g., the poisoned model is inaccurate for certain inputs (called subpopulation). The authors define model-targeting attack, in which the attacker aims to poison training data such that the learnt model is a given target model. The authors define distance between the poisoned model and the target model using their loss difference (this reduces to objective driven poisoning attack). Evaluation on linear SVM is performed and comparison with one existing attack is conducted.

1. The model-targeted poisoning attack is an ill-defined problem. When measuring the model-targeted poisoning attack, one should use some distance between the poisoned model and the target model, e.g., L2 distance (it's good that the evaluation considers this). The loss based metric reduces the model-targeted poisoning attack to be similar to objective-drive attack. However, when measuring success using distance between the poisoned model and the target model, the attack is obviously unsuccessful when the loss function is non-convex. Many ML loss functions are non-convex. In such non-convex cases, you cannot get the same model parameters in multiple runs of the same algorithm and training data, even if there are no poisoned data points. So the attack could only be successful for strongly convex loss functions, which have a global optimal solution. I would suggest the authors to explicitly mention that the attack is limited to such setting. Moreover, redefine the success metric to measure the attack. However, once limited to such setting, the paper's contribution is also limited.

2. Insufficient evaluation. The evaluation does not compare with subpopulation attacks. I understand you study model-targeted attacks. Since the evaluation is for subpopulation attacks. It is still interesting to know the comparison results.

3. Only linear SVM is evaluated. I suggest evaluating other models and datasets.

4. Why is model-targeted poisoning attack relevant? Objective-driven attacks are more relevant. In fact, the evaluation is on subpopulation attack, which is an objective-driven attack.


Typos

is the setting use in many prior works -- used

also allow arbitrary selectiong of the poisoning points -- selection

classifier that that has 0% accuracy -- that

---

> ### Author Response · Authors · 2020-11-16
> **Individual Response to AnonReviewer 4 (Part 1)**
>
> > 1. The model-targeted poisoning attack is an ill-defined problem. When measuring the model-targeted poisoning attack, one should use some distance between the poisoned model and the target model, e.g., L2 distance (it's good that the evaluation considers this). The loss based metric reduces the model-targeted poisoning attack to be similar to objective-drive attack. However, when measuring success using distance between the poisoned model and the target model, the attack is obviously unsuccessful when the loss function is non-convex ... In such non-convex cases, you cannot get the same model parameters in multiple runs of the same algorithm and training data, ... So the attack could only be successful for strongly convex loss functions, which have a global optimal solution. I would suggest the authors to explicitly mention that the attack is limited to such setting. Moreover, redefine the success metric to measure the attack. However, once limited to such setting, the paper's contribution is also limited.
>
>
> We believe the model-targeted poisoning attack is well defined and well motivated, both in previous works and in our work, and provide more detailed responses to the specific issues below. (In this response, parenthetical citation (as in ICLR format) indicates the references are in the paper and numeric citation indicates references are not in the original submission.)
>
>   _"Use some distance metric"_ -- To prove our results in the most general form, we used the loss based distance. We think the loss based distance is by itself an important distance because it captures the semantic closeness (i.e., similar prediction performance, which matters most in practice) of two models, instead of just the parameter closeness. This distance can also be related to other distances such as $\ell_1$. In particular, for the case of Hinge loss, we prove this relation in Appendix B and show that minimizing the loss based distance implies minimizing the model $\ell_1$ distance. Our empirical results also confirm the monotonic relationship of the two metrics for Hinge loss.
>
>   _"non-convex loss"_ -- Our convergence theorem does require convexity of the loss function. Poisoning attacks for convex models are hard problems, and previous poisoning attacks with theoretical guarantees also require the convexity of the loss function. Our approach is a significant advance compared to previous results because 1) it can be made to fit different attacker objectives (subpopulation and indiscriminate settings are demonstrated in the paper), in contrast to the previous attack with theoretical guarantee that only applies to indiscriminate setting (Steinhardt et al., 2017); and more importantly, 2) our attack has provable convergence compared to the state-of-the-art model-targeted attack, which has no such guarantees (Koh et al., 2018).
>
>   As mentioned in our contributions section (Section 1), our convergence property requires convex loss and strongly convex regularizer, which is common for convex models deployed in practice. Studying attacks on convex models is the important step towards understanding poisoning attacks for both convex and non-convex models. Many online learning algorithms for non-convex loss functions are also studied [1], [2] and we believe our framework can be a good starting point for applying these new online learning techniques to attack non-convex models. Also, note that, many real-world applications of machine learning deal with convex models and it is also important to understand this setting fundamentally.
>
>   [1] Xiang Gao et al., "Online Learning with Non-Convex Losses and Non-Stationary Regret", AISTATS 2018.
>
>   [2] Omar Besbes et al., "Non-stationary Stochastic Optimization", Operations Research, 2015.
>
>
>   _"redefine success metric"_ -- Sorry, we couldn't fully understand this comment and would appreciate more clarification if we are misunderstanding the comment. Based on our interpretation, we assume the reviewer is asking if we used the right metric to evaluate our attack. In our evaluation, we consider two metrics, that are 1) the convergence to the target model (in the loss based distance and actual model-distance) and 2) attack success in achieving the attacker objective encoded in the target model.
>
>   The _convergence metric_ is important as our model-targeted attacks try to get close to a target model, which encodes a certain attacker objective. This way, we can verify our theoretical results about the convergence of our attack. The _attack success_ metric is also important because target models encode certain attacker objectives and we need to ensure that inducing models closer to the target models can also achieve the same encoded attacker objectives. This measures if our attack is useful in practice. For details on why model-targeted attacks are relevant and why these attacks care about attacker objectives, please refer to our response to comment 4.

---

> ### Author Response · Authors · 2020-11-16
> **Individual Response to AnonReviewer 4 (Part 2)**
>
> > 2. Insufficient evaluation. The evaluation does not compare with subpopulation attacks. I understand you study model-targeted attacks. Since the evaluation is for subpopulation attacks. It is still interesting to know the comparison results.
>
> As mentioned in Section 5 (the experiment section), to compare with objective-driven attacks (of certain attacker objectives) fairly, we need to find target models that can achieve the attacker objectives with fewer poisoning points using our attack.
>
> Since the systematic search of such target models is an independent research question and is out of the scope of this paper, we did not compare to objective-driven attacks in the original submission. However, for the indiscriminate setting, some empirical evidence indicates that, generating target classifier with lower loss on the clean training set and higher error rate (than what is needed in the attacker objective) will need fewer poisoning points to achieve the attacker objective. We believe similar logic may also apply for the subpopulation setting and we are currently in the process of comparing to the subpopulation attack used by Jagielskiet et al. (2019). We will update the paper ("Version 3") once the results are ready.
>
> We also clarify that, in addition to the subpopulation setting, we also evaluated our attack in the conventional indiscriminate attack scenario, but deferred the details to Appendix D due to space limitation.
>
> - **Version 3 Update:** results of comparison to the subpopulation attack are now available in the updated paper.
>
> > 3. Only linear SVM is evaluated. I suggest evaluating other models and datasets.
>
> We are in the process of running additional experiments and will update the paper with new results ("Version 3").
>
> - **Version 3 Update:** new results on additional model and dataset are now available in the updated paper.
>
> > 4. Why is model-targeted poisoning attack relevant? Objective-driven attacks are more relevant. In fact, the evaluation is on subpopulation attack, which is an objective-driven attack.
>
> We clarify that the end goal of model-targeted attack is still to achieve certain attacker objectives efficiently (i.e., using fewer number of poisoning points) because the target models encode those attacker objectives.
>
> Specifically, model-targeted attacks consist of two parts: 1) generating a target classifier that encodes a certain attack objective (e.g., increase error on subpopulation), and 2) generating poisoning points to induce the target classifier. Our paper, along with previous model-targeted poisoning attacks, focuses on solving part 2, using a variety of target models to test our attack. Part 1 itself is an independent research question and deserves more investigation in the next step as to how to generate target models that can be induced more efficiently (with fewer poisoning points) while achieving a given attacker objective.
>
> At a high level, for model-targeted attacks, decomposing into two parts enables us to analyze different attacker objectives in a unified poisoning framework, as long as the target models of different attacker objectives (indiscriminate and subpopulation scenarios are demonstrated in the paper) can be generated properly.
>
> Additionally, objective-driven attacks can be used to generate a target model, which can then be used as the target for a model-targeted attack, resulting in an attack that achieves the desired attacker objective with fewer poisoning points. For example, in our experiments, the target classifiers are generated from the label flipping based objective driven attacks that are effective but need too many poisoning points to achieve their objective. Then, our attacks are deployed to achieve the same objective with fewer poisoning points.
>
> For example, in the indiscriminate attack on MNIST 1-7 dataset, using our model-targeted attack with a target classifier of 5\% error rate (generated by label-flipping attacks), reduces the number of poisoning points needed by 73\% (see the updated Appendix E in the revised version ("Version 2") for more details). In the indiscriminate setting, our attack (with classifiers from label-flipping attacks) also outperforms the state-of-the-art min-max attack (Steinhardt et al., 2017) at reducing the overall accuracy, with the same number of poisoning points. For example, the clean accuracy of SVM model on MNIST 1-7 is 98.9\% and the min-max attack reduces the accuracy to 93.9\% at 15\% poisoning ratio (650 poisoning points) while our attack reduces to 88.6\%. More results on different poisoning ratios can be found in the updated Appendix E in the revised paper ("Version 2").

---

### Official Review · AnonReviewer1 · 2020-10-28
**Review for Model-Targeted Poisoning Attacks with Provable Convergence**

**Rating:** 7
**Confidence:** 5

**Review:**

The paper studies model-targeted poisoning attacks during training time. The victim learner is assumed to be a convex learner that performs regularized empirical risk minimization. Instead of formulating the problem as a bi-level optimization as standard, the paper proposes reducing the attack to an online learning problem, where the adversary decides on a poisoning data point in a sequential manner. The data point that leads to the largest difference of loss between the current model and the target model is selected as the next point to inject into the training data. In doing so, the learned model will converge to the target model in the long run. The paper provided a theoretical analysis on how many data points are needed to achieve a pre-specified small distance to the target model. Experiments demonstrated the effectiveness of the proposed attack.

The main advantage of the paper lies in that it draws the first connection between the data poisoning attack and online learning. The techniques used in the paper very novel. While the proposed attack algorithm looks simple, it actually has a deeper theoretical reason for why the attack can be effective. The authors also give a rigorous theoretical result on how many poisoning points are needed in order to drive the learner’s model toward a target model. This result is elegant!

Another strength of the paper is that the proposed online learning-based attack can even outperform the traditional KKT attacks. This is surprising to me because the KKT formulation is in fact an optimal-attack formulation. The only place that could incur sub-optimality in the KKT approach is the nonlinearity of the attack optimization, which may not solve to the global-optima solution. I am wondering if the authors could provide some intuition about why the proposed attack results in superior attack performance as compared to traditional data poisoning attacks?

I am also curious if the attack in this paper extends to the deep learning setting, where the victim learners are no longer convex. Empirically it would be nice to show some results on that, although the theoretical results definitely do not easily extend. Currently, the paper only experiments on simple tasks where the dataset is kind of small. It would be more convincing to study larger datasets and verify whether the proposed attack can outperform existing baselines across a variety of tasks.

Overall, I enjoy reading the paper, and I think it makes significant contributions that push forward the frontier of data poisoning attack. In particular, the online learning approach is very appealing and gives a very nice formulation of the model-targeted attack.

---

> ### Author Response · Authors · 2020-11-16
> **Individual Response to AnonReviewer1**
>
> > 1. Another strength of the paper is that the proposed online learning-based attack can even outperform the traditional KKT attacks. This is surprising to me because the KKT formulation is in fact an optimal-attack formulation. The only place that could incur sub-optimality in the KKT approach is the nonlinearity of the attack optimization, which may not solve to the global-optima solution. I am wondering if the authors could provide some intuition about why the proposed attack results in superior attack performance as compared to traditional data poisoning attacks?
>
> The foundation of the KKT attack is that, for binary classification, any target classifier generated by training on set $\mathcal{D}$ with size $n$, the (exact) same classifier can also be obtained by training on set $\mathcal{D}^{'}$ with size at most $n$ and the set only contains two distinct points, one from each class. In practice, KKT attack often aims to induce the exact same classifier with (much) smaller number of poisoning points, which may not be feasible and KKT attack may fail in these situations.
>
> Our attack does not try to obtain the exact classifier and only tries to approach it with the minimum number of poisoning points. That is why we can get close to the target classifier with fewer poisoning points than the number of points used to exactly produce the target classifier. This is not guaranteed, however, and sometimes does not happen. For example, in the indiscriminate attack scenario (Appendix D), the KKT attack sometimes slightly outperforms our attack.
>
> > 2. I am also curious if the attack in this paper extends to the deep learning setting, where the victim learners are no longer convex. Empirically it would be nice to show some results on that, although the theoretical results definitely do not easily extend. Currently, the paper only experiments on simple tasks where the dataset is kind of small. It would be more convincing to study larger datasets and verify whether the proposed attack can outperform existing baselines across a variety of tasks.
>
> This comment is related to comment 1 from AnonReviewer 2, please see our response there. We agree that having results on larger models and datasets will be valuable, but are not sure if we can have the results ready before the rebuttal deadline.

---

### Official Review · AnonReviewer3 · 2020-10-28
**An online algorithm for targeted poisoning, with theoretical guarantees**

**Rating:** 6
**Confidence:** 3

**Review:**

**Paper summary**
The paper proposes an algorithm, that works in an online fashion, for targeted poisoning attacks. If the loss function is convex, then the algorithm is guaranteed to converge to the target as the number of poisoned samples increases. The paper claims that this is the first model-targeted attack which has theoretical guarantees. The lower bound provided is interesting in the sense that it can give a lower bound on the number of samples needed to reach the target model from the current model.

**Strengths**
1. The paper proposes a new algorithm that works by progressively adding poisoned points to the dataset. Theoretical guarantee (Theorem 4.1) is also provided for the algorithm. The proof idea is well explained and I liked the connection with online learning and how the algorithm is reduced to follow-the-leader algorithm.
2. The algorithm works by progressively adding more points and hence it can stop as soon as it is close to the target. This means that it does not need a predecided budget and if it is indeed optimal, then it would use the minimum number of poisoned points to reach the target.
3. On SVMs, the experiments show that the attack is almost optimal in the sense that it matches the theoretical lower bound.

**Concerns**
1. Theorem 4.2 is a lower bound for the minimum number of samples needed to reach the target model exactly. Can we say something about reaching the target model approximately (say up to distance $\epsilon$)? I think this can be achieved by the following optimization in Theorem 4.2: $$\inf_{\theta':\|\theta'-\theta_p\|\leq\epsilon} \sup_\theta c(\theta,\theta')=\frac{L(\theta';\mathcal{D}_c)-L(\theta;\mathcal{D}_c)+NC_R(R(\theta_p)-R(\theta))}{\sup(l(\theta;x,y)-l(\theta';x,y))+C_R(R(\theta)-R(\theta')))}.$$

Further, how much would the lower bound decrease (as a function of $\epsilon$) if we are indeed interested in only reaching the target model approximately?

2. In the experiments, the lower bound is computed for the number of samples needed to reach the model induced after poisoning. This can be different from the target model. Hence, shouldn't the lower bound be computed for the number of samples needed to reach the target model? I think that would be the true lower bound for poisoning. However, I believe that the two numbers should be close.

3. The experiments are performed only for linear SVMs. It would be more interesting to see if the attack also works well on deep neural networks. This is also important because the paper claims that the existing algorithms get stuck in bad local minima. Thus, it should be checked if this algorithm also gets stuck in bad local minima.

**Score justification**

Model poisoning is a very real threat in modern machine learning. Strong attacks provide good insights for developing strong defenses. I believe the attack proposed in this paper is strong and theoretically backed. Further, the paper also provides lower bound on the minimum amount of poisoning needed to reach the target model. However, I have some concerns regarding the attack's success on deep neural networks.

---

> ### Author Response · Authors · 2020-11-16
> **Individual Response to AnonReviewer3**
>
> > 1. Theorem 4.2 is a lower bound for the minimum number of samples needed to reach the target model exactly. Can we say something about reaching the target model approximately (say up to distance
> $\epsilon$)? I think this can be achieved by the following optimization in Theorem 4.2: ... Further, how much would the lower bound decrease (as a function of $\epsilon$) if we are indeed interested in only reaching the target model approximately?
>
> The formulation given by the reviewer is correct, where $\epsilon$ is defined with respect to loss-based distance. However, if we want to actually compute such a value, we need to solve a bi-level optimization problem which might not be feasible in general settings. Therefore, we need some further restriction for obtaining such a lower bound for classifiers that are $\epsilon$-close to the target model. For example, we can derive such a bound if the loss based distance is bi-directional, meaning that if $\theta_1$ is $\epsilon$-close to $\theta_2$ then $\theta_2$ is also $O(\epsilon)$-close to $\theta_1$. Indeed, we believe this is the case for the Hinge loss and we can derive a lower bound for it. We will make this clear in the next round of updated paper ("Version 3").
>
> Compared to the lower bound of the target model, the lower bound for $\epsilon$-close models (to the target model) will always be smaller, but the gap shrinks as $\epsilon\rightarrow 0$.
>
> - **Version 3 Update:** related corollary and the corresponding proof are now available in the updated paper.
>
>
> > 2. In the experiments, the lower bound is computed for the number of samples needed to reach the model induced after poisoning. This can be different from the target model. Hence, shouldn't the lower bound be computed for the number of samples needed to reach the target model? I think that would be the true lower bound for poisoning. However, I do believe that the two numbers should be close.
>
> Our lower bound is for achieving the exact target classifier, hence we compute the lower bound for model induced from our poisoning. This way, we know how optimal our attack is in inducing the poisoned model.
> For the actual target model, we can still compute the lower bound. However, the lower bound is for achieving the exact model and hence, it does not show the tightness of our attack, because our attack never gets to the exact target model and only gets close to it.
>
> > 3. The experiments are performed only for linear SVMs. It would be more interesting to see if the attack also works well on deep neural networks. This is also important because the paper claims that the existing algorithms get stuck in bad local minima. Thus, it should be checked if this algorithm also gets stuck in bad local minima.
>
> This comment is similar to comment 1 from AnonReviewer 2, please see our response there. We are also in the process of testing our attack empirically on deep neural networks, but not sure if we can have the results ready before the rebuttal deadline.
>
> Our statement that _gradient based attacks can get stuck into bad local optima_ reiterates claims from the previous works on poisoning attacks by Steinhardt et al. (2017) and Koh et al. (2018) (references in the paper), in which they empirically found that gradient attacks empirically underperform the min-max attack and the KKT attack.
>
> The statement means that the issues with local optima exist when the victim models are convex (e.g., SVM). A poisoning attack is in essence a bi-level optimization problem, which is hard to solve in general, even for convex loss functions. Gradient attacks leverage the KKT condition to approximately convert the bi-level optimization problem into a simple optimization problem. Because an approximation is used and gradient based local optimization techniques are deployed to solve the approximated problem, there exists the issues of local optima when attacking the convex models.

---

### Official Review · AnonReviewer2 · 2020-11-05
**Interesting problem but weak evaluation**

**Rating:** 5
**Confidence:** 4

**Review:**

This paper presents an improvement on an interesting problem: poison a dataset to induce a machine learning process to a misleading model. This field of study has been trending in recent years, and this paper presents a neat increment over previous works. The basic idea is that the attacker iteratively generates new data points so as to minimize the difference between the poisoned model and the target model. This approach requires retraining the model in each iteration, which is a costly procedure. This work then propose to use an online learning approach to update the model with a small cost to accelerate the entire process, and thus provide an efficient data poisoning approach.

There are several issues with both the work itself and the presentation. The main issue is about the evaluation. The approach is only evaluated on one dataset (Adult dataset), considers only one model (SVM) and compared against one approach (KKT attack). This is not enough to justify the universal effectiveness of the approach applying to other machine learning algorithms. In particular, modern deep learning algorithms typically suffer catastrophic forgetting issues when applying online learning algorithms, and these models are of more interests in the context of poisoning attacks. So this work does not provide enough evidence to justify that the proposed approach is effective in dealing with them.

Maybe I neglect them, but I do not find the description of the adult dataset's stats, i.e., how large is the dataset? The paper studies poisoning more 1500 data points, what percentage is these data to the entire dataset? It looks like, from the description, that these dataset might be 100% of the entire dataset. Isn't this too large to be feasible in the real setup?

In addition, it might be important to compare against Jagielski's 2018 paper [1], which is not cited, as well in the non-subpopulation setup, in addition to KKT, which should provide a better baseline. This work also presents a set of commonly used dataset and machine learning models, which can be experimented with in the context of this work.

Part of the above is presentation issues, and there are more issues in the presentation of the theoretical part. For example, the presentation of Thm 4.1 uses the term of epsilon-close, but I don't find its formal definition above (like in Sec 2). Thm 4.2 uses a notion of c(theta) without a definition of what it is. It seems c indicates a clean dataset (when in subscription), but I find it difficult to read it when there is an argument theta associated with it. These issues make me hard to understand what the main theorems of this work want to tell us.

I think the paper should be revised to fix the presentation issues first so that we can evaluate more accurately whether the theoretical results have enough merit to upgrade my rating; also it may be helpful if more experiments can be conducted to demonstrate that the approach is effective on more datasets and more machine learning models.

[1] Matthew Jagielski, et al. Manipulating machine learning: Poisoning attacks and countermeasures for regression learning, SP 2018

---

> ### Author Response · Authors · 2020-11-16
> **Individual Response to AnonReviewer2**
>
> > 1. The main issue is about the evaluation. The approach is only evaluated on one dataset (Adult dataset) ... This is not enough to justify the universal effectiveness of the approach ...
>
> To clarify, in addition to our theoretical results (which do not depend on the dataset), and the results for Adult in the main body of the paper, we also evaluated our attack on the MNIST 1-7 dataset. The MNIST 1-7 results are in Appendix D, but were not included in the main body of the paper due to space limitations. MNIST 1-7 is widely used in evaluating previous poisoning attacks on classification problems.  We are currently running additional experiments and will update the paper (forthcoming "Version 3") when the results are ready.
>
> It is true that our results are limited to convex models, but this reflects the current state-of-the-art in this research area. For deep learning models or more generally, non-convex models, we cannot establish a convergence guarantee as the assumption of convex loss function is violated. This is in line with all previous work on poisoning attacks, none of which have provided any convergence guarantees for non-convex learning. We believe our exploration of convex models is an important step towards understanding the poisoning attacks and is a step towards understanding attacks for non-convex models. Incorporating online learning frameworks for non-convex loss functions into our attack could be one possible path.
>
> > 2. ... I do not find the description of the adult dataset's stats, i.e., how large is the dataset? ... Isn't this too large to be feasible in the real setup?
>
> We downsampled the Adult dataset to ensure it is class-balanced and we ended up having 15,682 training and 7,692 test data (1,500 poisoning points of subpopulation attack occupy around 10\% of the entire training set size). Details of the dataset description are in Appendix D in the original submission. In the (now available) revised version ("Version 2"), we have added a description of the dataset at the beginning of Section 5 in the main body.
>
> Across the paper, we have results for poisoning ratios ranging from $\approx 5\%$ to $\approx 50\%$. In our threat model, the adversaries are interested in inducing target models that encode certain attacker objectives (e.g., 0\% accuracy on the subpopulation). Therefore, if the encoded attacker objective is hard (e.g., significant reduction in the accuracy), then inducing the corresponding target model also requires a larger number of poisoning points. The experiments in the paper are designed to evaluate how efficiently our attack can induce any given target model (and hence the encoded attacker objective) which is why we include large poisoning ratios in the experiments, although we agree that such attacks are not likely to be relevant in practice.
>
> > 3. In addition, it might be important to compare against Jagielski's 2018 paper [1] ...
>
> In the revised paper ("Version 2"), we have added a discussion about the referred paper in the related work section (Section 3). Note that the referenced paper focuses on regression tasks while our paper mainly focuses on classification problems.
>
> > 4. ... there are more issues in the presentation of the theoretical part. For example, the presentation of Thm 4.1 uses the term of epsilon-close, but I don't find its formal definition above (like in Sec 2) ...
>
> In the (original) paper, we define our loss-based distance right before the statement of Theorem 4.1. We think the confusion arises from the fact that we did not define the term $\epsilon$-close in Definition 1. Model $\theta_1$ is $\epsilon$-close to model $\theta_2$ means the loss-based distance (given in Definition 1 in the main body) from $\theta_1$ to $\theta_2$ is upper bounded by $\epsilon$. In the revised paper ("Version 2"), we have clarified the meaning of $\epsilon$-close in Definition 1.
>
> In fact, the notion $c(\theta)$ is completely unrelated to the subscript $D_c$ which refers to the clean set. We agree that $c$ is overloaded here. The notation $c(\theta)$ was intended to denote the large formula after the supremum so that we can refer to it easily in the proof. In the revised paper (i.e., Version 2), we have changed the notation to $z(\theta)$ across the paper to make this clear.
>
> > 5. I think the paper should be revised to fix the presentation issues first so that we can evaluate more accurately whether the theoretical results have enough merit to upgrade my rating; also it may be helpful if more experiments can be conducted to demonstrate that the approach is effective on more datasets and more machine learning models.
>
> In our revised version ("Version 2"), we have fixed the mentioned presentation issues, and we appreciate your willingness to reconsider our theoretical results.
>
> We are in the process of running additional experiments and will update the paper ("Version 3") once new results are ready.
>
> - **Version 3 Update:** new results are now available.

---

### Author Response · Authors · 2020-11-16
**General Response to Reviewers**

We have uploaded a revised paper that fixes presentation issues and we refer to it as "Version 2". In Version 2, the changes are (pointers to the changes in the paper are also given in the individual comment responses):

- A description of Adult dataset is added to the beginning of Section 5 (the experiment section, in response to comment 2 from AnonReviewer 2);
- A discussion on the referenced paper by Jagielski et al. (2018) is added to the related work section (section 3, in response to comment 3 from AnonReviewer 2);
- Clarification of $\epsilon$-close in Definition 1 in the main body, and the new notation $z(\theta)$ for the (original) confusing notation $c(\theta)$ in Theorem 4.2, and the proof of Theorem 4.2 in Appendix A (section 4.2, in response to comment 4 from AnonReviewer 2);
- The Appendix E is updated (from the original version) to include a more detailed comparison to objective-driven attacks (in response to comment 4 from AnonReviewer 4);
- Typos pointed out by AnonReviewer 4 are fixed;

Before the rebuttal deadline, we will post another revision that includes the results of additional experiments and formal proofs of an additional question raised by reviewer 3. We wanted to post the presentation revisions and comment responses now to have some time for discussion while we are still working on completing the additional experiments for the second revision. In our comment responses, we refer to the revision that has been uploaded now as "Version 2", and the future revision that will be posted later as "Version 3".

---

### Author Response · Authors · 2020-11-24
**Version 3 is Available with Additional Experiments and Proofs**

We have uploaded the revised paper (Version 3) that includes the results of the additional experiments and some formal proofs. A special note (**Version 3 Update**) is also added to each individual comment that is related to Version 3 updates. In Version 3, the changes are:

- In response to Comment 5 from AnonReviewer 2, Comment 3 from AnonReviewer 4, we have added a new Appendix F that reports on our new experiments evaluating our attack on an additional dataset and model. We used the Dogfish dataset, which is also used for evaluating poisoning attacks by previous works  (Koh & Liang, 2017; Steinhardt et al., 2017; Koh et al., 2018) (references in the paper). We also evaluated our attack on a logistic regression model, which is also adopted in the previous work (Mei & Zhu, 2015b) (reference in the paper). All the new results (SVM on Dogfish, logistic regression on Adult, MNIST and Dogfish) confirm the effectiveness of our attack and that it outperforms the state-of-the-art KKT attack.

- In response to Comment 2 from AnonReviewer 4, we updated Appendix E (in Version 2) and added the comparison to the subpopulation attack adopted by Jagielskiet et al. (2019) (reference in the paper), for both the SVM and logistic regression model. The results confirm the superior performance of our attack against the objective-driven attacks in the subpopulation setting.

- In response to Comment 1 from AnonReviewer 3, we added a corollary (Corollary 4.2.1 in Version 3) to Theorem 4.2 in Section 4 in the main body to mention the lower bound for models that are $\epsilon$-close to the target model. We added a formal proof of this result to Appendix A (right below the proof of Theorem 4.2). We added a definition of bi-directional closeness (required for proving the lower bound for $\epsilon$-close models) to the explanation of Definition 1 (right below the Definition 1). A formal proof that Hinge loss satisfies the bi-directional closeness is included in Appendix B (Corollary B.2.1 in Version 3).

- Section 5 (experiment section) in the main body is slightly updated to reflect the inclusion of new results.

We are still in the process of testing on deep learning models and unfortunately, results will largely not be ready before the rebuttal deadline. As mentioned in the previous responses, however, we feel our results for convex learning are a significant scientific advance and consistent with previous evaluations in this area. If our response leaves questions open, we are happy to discuss the issues further.

---

### Decision · Program_Chairs · 2021-01-07
**Final Decision**

**Decision:**

Reject

**Comment:**

The paper establishes an interesting relationship between poisoning and online learning. Instead of framing the poisoning problem as a bi-level optimization problem as what is done conventionally, the paper proposes reducing the poisoning attack design to an online learning problem in which the adversary decides on a poisoning data point in a sequential manner.  The data point that leads to the largest difference of loss between the current model and the target model is selected as the next point to inject into the training data.

Pros
+ If the loss function is convex, the proposed algorithm is guaranteed to converge to the target, and the paper provides a lower bound on the number of samples needed to reach the target model from the current model.

+ Experiments on SVMs show the advantageous performance of the proposed attack algorithm.


Cons
- The reason why the proposed online learning method could outperform the KKT-based attack is not well justified theoretically. Experimentally, the paper would be stronger if evaluations are done on more diverse settings and data.

- Reviewers have expressed concerns on how practical the proposed algorithm is, since the guarantees are established on convex loss functions. The paper would be stronger if  the authors can further compare the attack with deep-learning poisoning algorithms on larger datasets.


I truly believe that the paper’s exploration of convex models is an important step towards understanding the poisoning attacks and is a step towards understanding attacks for non-convex models. However, I do think that the paper could make a more profound contribution and impact with stronger experimental evaluations.

Therefore, I would classify this paper as borderline, toward weak reject compared with other papers.